# MESSAGE FUNCTION SEARCH FOR HYPER-RELATIONAL KNOWLEDGE GRAPH

## ABSTRACT

Recently, the hyper-relational knowledge graph (HKG) has attracted much attention due to its widespread existence and potential applications. The pioneer works have adapted powerful graph neural networks (GNNs) to embed HKGs by proposing domain-specific message functions. These message functions for HKG embedding are utilized to learn relational representations and capture the correlation between entities and relations of HKGs. However, these works often manually design and fix structures and operators of message functions, which makes them difficult to handle complex and diverse relational patterns in various HKGs (i.e., data patterns). To overcome these shortcomings, we plan to develop a method to dynamically search suitable message functions that can adapt to patterns of the given HKG. Unfortunately, it is not trivial to design an expressive search space to enable the powerful message functions being searched, especially the space cannot be too large for the sake of search efficiency. In this paper, we first unify a search space of message functions to search both structures and operators. The message functions of existing GNNs and some classic KG/HKG models can be instantiated as special cases of the proposed search space. Then, we leverage a search algorithm to search the message function and other GNN components for any given HKGs. We empirically show that the searched message functions are data-dependent, and can achieve leading performance in link/relation prediction tasks on benchmark HKGs.

## 1 INTRODUCTION

Knowledge base (KB) (Auer et al., 2007) is an important tool to organize and explore human knowledge, thereby promoting a series of applications, e.g., question answering (Lukovnikov et al., 2017) and recommendation system (Cao et al., 2019). Generally, the KB stores the $n$-ary fact $r(e_1, \cdots, e_n)$ ($n$ is arity) that represents the relation $r \in R$ between real-world entities $e_i \in E$. To manipulate large scale KBs, KB embedding (Nickel et al., 2015; Wang et al., 2017) proposes to encode the set of relations $R$ and entities $E$ into a $d$-dimensional vector space $\boldsymbol{R} \in \mathbb{R}^{|R| \times d}$, $\boldsymbol{E} \in \mathbb{R}^{|E| \times d}$. In last decades, the research community mainly focuses on embedding knowledge graphs (KGs) that only contain binary facts $\{r(e_1, e_2)\}$, e.g., `isCaptialOf(Beijing,China)`. Among kinds of KG embedding models (Rossi et al., 2021), tensor models (Lacroix et al., 2018; Balazevic et al., 2019) propose to represent a KG into a 3-order tensor and decompose tensors into $\boldsymbol{R}$ and $\boldsymbol{E}$, which achieve outstanding empirical performance and theoretical guarantees. Recent studies start to learn embeddings from $n$-ary facts ($n \geq 2$) because $n$-ary facts are widespread in KBs, e.g., `playCharacterIn(LeonardNimoy,Spock,StarTrek1)`. For example, more than 30% of entities in Freebase (Bollacker et al., 2008) involve facts with higher arity (Wen et al., 2016). Therefore, it is necessary to investigate the more general case of KGs, facts with mixed arities $S = \{r(e_1, \ldots, e_n) : n \in \{2, \ldots, N\}\}$ named as hyper-relational KGs (HKGs).

Unfortunately, it is hard to extend powerful tensor models from the case of fixed arity (e.g., KG) to the case of mixed arities (i.e., HKG). That is because a tensor can only model a set of facts under the same arity. Instead, some pioneer works (Yadati, 2020; Galkin et al., 2020) demonstrate that the multi-relational hypergraph (MRHG) could be a more natural way to model HKGs (see Appx. A for more details). Let entities $E$ and relations $R$ be nodes and edge types in the MRHG $\mathcal{G}(E, R, S)$, respectively. The length of MRHG's edge $(e_1, \ldots, e_n)$ (hyperedge) could be variant, which can represent facts with various arities $n$. And hyperedges can be labeled by multiple edge types $r \in R$ like $r(e_1, \ldots, e_n) \in S$. Under the MRHG modeling, these works adapt powerful graph

neural networks (GNNs) (Kipf & Welling, 2016; Hamilton et al., 2017) to embed HKGs (Yadati, 2020; Galkin et al., 2020). Generally, GNNs learn node embeddings by passing messages (e.g., the node features) from adjacent nodes to the center node. But in scenarios of HKGs, it is important to know the type of edge (relation) that connects several entities. Therefore, existing GNNs for HKG embedding design several domain-specific message functions to learn relational representations $\boldsymbol{R}$ by capturing the interaction between entities $E$ and relations $R$.

Existing works manually design and fix the structures and operators of message functions. However, such rigid message function designs are not conducive to pursuing high empirical performance, as relations usually have distinct patterns in various KGs/HKGs. For example, the message function of G-MPNN (Yadati, 2020) adopts the inner product way like DistMult (Yang et al., 2015) to compute the correlation between entities and relations, which has been proven to only cover symmetric relations (Kazemi & Poole, 2018). Its performance may not be good if there are many non-symmetric relations existed (see Appx. B.1). It may be a potential solution to design a universal message function to cover as many relational patterns as possible. But covering a certain pattern does not mean that the model can reach good performance on it (Meilicke et al., 2018; Rossi et al., 2021). Moreover, a pioneer work AutoSF (Zhang et al., 2020) shows that designing data-aware models can consistently achieve high performance on any given KGs. Thus, dynamically searching message functions could be an effective way to capture the data patterns of the given HKG and pursue high performance.

Unfortunately, the searching method AutoSF is strictly restricted to bilinear KG models (Yang et al., 2015; Kazemi & Poole, 2018), which is not applicable to message function design and the HKG scenario. Besides, although neural architecture search (NAS) (Elsken et al., 2019) has been introduced to search GNN architectures, current GNN search spaces (Zhang et al., 2021) are not well regularized to tasks on HKGs. Specifically, the existing search spaces of message function follow the classic way to simply aggregate node neighbors to learn node embeddings, while ignoring edge embeddings to represent relations. Thus, they cannot capture the correlation between entities and relations of HKGs.

In summary, rigid message function designs for HKGs are not conducive to consistently pursuing high performance on different data sets, while existing searching methods are not applicable for HKGs. To bridge this research gap, we propose the *M*essage function *SEA*rch for any given *HKG*s, named as MSeaHKG. However, it is non-trivial to design an expressive search space to enable the powerful message functions being searched, especially the space cannot be too large for the sake of search efficiency. Thus, we identify the necessary computation operators that are domian-specific designs for HKGs and propose to search the interaction between these operators for capturing the relational patterns. Moreover, except for the message function search, we also incorporate other GNN components (e.g., aggregation function) in the MSeaHKG search space for more performance improvements. Then, we formulate the discrete HKG models with probabilistic modelings to enable an efficient NAS algorithm working on our scenario. The main contributions are listed as:

- Previous GNN searching methods generally ignore the edge representations, which fails to handle semantic meaningful relations on HKGs. Besides, their message functions cannot capture complex interactions between entities with relations. In this paper, we propose a searching method to dynamically design a suitable GNN that can achieve high performance on the given HKG.

- Inspired by rigid message function designs, we define a novel search space of message functions for HKGs, which enables the message function to be flexibly searched on the given HKG. Especially, the message function designs of existing GNNs for HKGs and some classic KG/HKG models can be instantiated as special cases of the proposed space.

- We conduct experiments on benchmark HKGs for the link prediction and relation prediction tasks. Experimental results demonstrate that MSeaHKG can consistently achieve state-of-the-art performance by designing data-aware message functions. Besides, we also transfer MSeaHKG to other graph-based tasks and further investigate its capability.

## 2 RELATED WORK

### 2.1 ONE-SHOT NEURAL ARCHITECTURE SEARCH IN GRAPH NEURAL NETWORK

To avoid manual efforts on neural architecture designs, NAS (Hutter et al., 2018; Yao & Wang, 2018) aims to automatically search suitable neural architectures for the given data and task. Generally, search

Table 1: Overview of Existing Works. NC, GC, LP, RP denote node classification, graph classification, link prediction, relation prediction, respectively. $DP(\cdot)$ is dropout; $BN(\cdot)$ is batch normalization; $\bar{o}(\cdot)$ outputs the hidden representation after summarizing mixed operations (Zhao et al., 2021); $\boldsymbol{h}_i^K$ denotes the output from $i$-th operator of $\mathcal{O}$ in $K$-th layer of message function (see Sec. 3.1).

| Type | Model | Scenarios | | Message Function |
|------|-------|-----------|---|------------------|
| | | Task | # Type/Length of edge | |
| **NAS for GNNs** | You et al. (2020) | NC/GC/LP | $= 1/= 2$ | $DP(BN(\boldsymbol{W}\boldsymbol{e}_j + \boldsymbol{b}))$ |
| | GraphNAS | NC | | $a_{ij}concat(\boldsymbol{e}_i, \boldsymbol{e}_j)$ |
| | AGNN | | | $a_{ij}\boldsymbol{W}\boldsymbol{e}_j$ |
| | SANE | | $\geq 1/= 2$ | $\boldsymbol{W}\bar{o}(\{\boldsymbol{e}_j\}_{e_j \in N(e_i)})$ |
| | NAS-GCN | GC | | $a_{ij}MLP(\boldsymbol{h}_{ij})\boldsymbol{e}_j$ |
| **GNNs for KGs/HKGs** | R-GCN | LP/RP | $\geq 1/= 2$ | $\boldsymbol{W}_r\boldsymbol{e}_j$ |
| | CompGCN | | | $\boldsymbol{W}_{\lambda(r)}\phi(\boldsymbol{e}_j, \boldsymbol{r})$ |
| | StarE | | $\geq 1/\geq 2$ | $\boldsymbol{W}_{\lambda(r)}\phi_r(\boldsymbol{e}_o, \gamma(\boldsymbol{r}, \boldsymbol{h}_r))$ |
| | G-MPNN | | | $\boldsymbol{r}_{P(e)} * \prod_i \boldsymbol{p}_{e_i} * \boldsymbol{e}_i$ |
| **Ours** | MSeaHKG | LP/RP | $\geq 1/\geq 2$ | $MLP\&concat(\{\boldsymbol{h}_i^K\}_{i=1}^{|\mathcal{O}|})$ |

space, search algorithm, and evaluation measurement are three important components in NAS (Elsken et al., 2019). Search space defines what network architectures in principle should be searched. The search algorithm performs an efficient search over the search space and finds architectures that achieve good performance. Evaluation measurement decides how to evaluate the searched architectures during the search. Classical NAS methods are computationally consuming because candidate architectures are evaluated by the stand-alone way, i.e., evaluating the performance of architecture after training it to convergence. More recently, one-shot NAS (Pham et al., 2018) proposes the weight sharing mechanism to share network weights across different candidate architectures and evaluate them on the shared weights, which can extremely reduce the search cost.

Some pioneer works have explored NAS for GNNs, such as You et al. (2020), GraphNAS (Gao et al., 2020), AGNN (Zhou et al., 2019), NAS-GCN (Jiang & Balaprakash, 2020). And the one-shot NAS also has been introduced to search GNN architectures recently, e.g., SANE (Zhao et al., 2021). As shown in the left part of Fig. 1, most GNN searching methods follow the message passing neural networks (MPNNs) (Gilmer et al., 2017) to unify two steps of the GNN framework in one layer:

$$step1: \boldsymbol{m}_i \leftarrow agg(\{mg_c(\boldsymbol{e}_i, \boldsymbol{e}_j)\}_{e_j \in N(e_i)}), \ step2: \boldsymbol{e}_i \leftarrow act(comb(\boldsymbol{e}_i, \boldsymbol{m}_i)), \quad (1)$$

where $\boldsymbol{e}_i \in \mathbb{R}^d$ represents the embedding of node $e_i$, $\boldsymbol{m}_i$ is the intermediate embeddings of $e_i$ gathered from its neighborhood $N(e_i)$. The search space of operators in Eq. 1 are summarized into:

- **Message Function** $mg_c(\cdot)$: The message function decides the way to gather information from a neighborhood $e_j$ of the center node $e_i$. Zhang et al. (2021) summarizes the typical message in existing GNN searching methods as $mg_c(\boldsymbol{e}_i, \boldsymbol{e}_j) = a_{ij}\boldsymbol{W}\boldsymbol{e}_j$, where $a_{ij}$ denotes the attention scores between nodes $e_i$ with $e_j$. Besides, we present more message function designs in Tab. 1.

- **Aggregation Function** $agg(\cdot)$: It controls the way to aggregate message from nodes' neighborhood. Usually $agg \in \{sum, mean, max\}$, where $sum(\cdot) = \sum_{e_j \in N(e_i)} mg_c(\boldsymbol{e}_i, \boldsymbol{e}_j)$, $mean(\cdot) = \sum_{e_j \in N(e_i)} mg_c(\boldsymbol{e}_i, \boldsymbol{e}_j)/|N(v)|$, and $max(\cdot)$ denotes channel-wise maximum.

- **Combination Function** $comb(\cdot)$: It determines the way to merge messages between neighborhood and node itself. In literature, $comb$ is selected from $\{concat, add, mlp\}$, where $concat(\cdot) = [\mathbf{e}_i, \mathbf{m}_i]$, $add(\cdot) = \mathbf{e}_i + \mathbf{m}_i$, and $mlp(\cdot) = MLP(\boldsymbol{e}_i + \boldsymbol{m}_i)$ ($MLP$ is Multi-layer Perceptron).

- **Activation Function** $act(\cdot)$: $[identity, sigmoid, tanh, relu, elu]$ are some of the most commonly used activation functions (Gao et al., 2020).

Overall, $mg_c(\cdot)$ in Eq. 1 only learns node embeddings, which cannot encode the semantic meaningful edge types (i.e., relations in HKGs). Note that NAS-GCN (Jiang & Balaprakash, 2020) takes the edge feature $\boldsymbol{h}_{ij}$ between $e_i$ and $e_j$ as input without learning edge embeddings, and recent AutoGEL (Zhili et al., 2021) simply extends the message function $mg_c(\cdot)$ to learn edge embeddings without studying the interactions between entities and relations, thereby failing to handle the LP/RP tasks on HKGs.

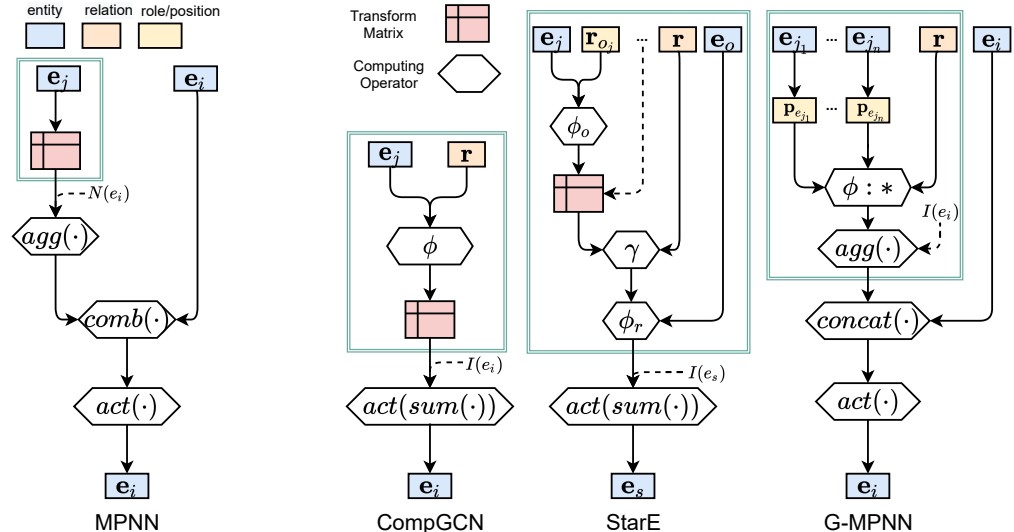

Figure 1: The framework of several GNN frameworks. The green box refers to the message function. Note that CompGCN and StarE select $sum(\cdot)$ as $agg(\cdot)$ and omit $comb(\cdot)$, while G-MPNN uses $concat(\cdot)$ as $comb(\cdot)$.

## 2.2 GRAPH NEURAL NETWORKS FOR KG/HKG EMBEDDING

As introduced in Sec. 1 and Sec. 2.1, message functions in classic GNNs simply aggregate messages from adjacent nodes (see Tab. 1 and Eq. 1). But in scenarios of KGs and HKGs, it is important to know the type of edge (relation) that connects several entities. To capture relations, R-GCN (Schlichtkrull et al., 2018) proposes to model relations $R$ in KGs through relation-specific weight matrix $\mathbf{W}_r \in \mathbb{R}^{d \times d}$ for $r \in R$ in message functions, which is instantiated as:

$$\boldsymbol{e}_i = act(sum(\{\boldsymbol{W}_r \boldsymbol{e}_j\}_{r(e_i, e_j) \in I(e_i)})), \tag{2}$$

where $I(e_i) = \{r(e_i, e_j) \in S : r \in R, e_j \in E\}$ is the set of relational edges incident on $e_i$. But such relation modeling may lead to the over-parameterization issue because there are many relations in KGs. Therefore, CompGCN (Vashishth et al., 2020) utilizes the embedding vector $\boldsymbol{r}$ to represent the relation instead of weigh matrix:

$$\boldsymbol{e}_i = act(sum(\{\boldsymbol{W}_{\lambda(r)} \phi(\boldsymbol{e}_j, \boldsymbol{r})\}_{r(e_i, e_j) \in I(e_i)})), \ \boldsymbol{r} = \boldsymbol{W}\boldsymbol{r} \tag{3}$$

where $\lambda(r)$ records the directional information of edges. The entity-relation composition operator set $\{sub, mult, corr\}$ is inspired by classical scoring function design in existing KG embedding models, such as element-wise subtraction $sub(\cdot) = \boldsymbol{e}_j - \boldsymbol{r}$ (Bordes et al., 2013), inner product $mult(\cdot) = \boldsymbol{e}_j * \boldsymbol{r}$ (Yang et al., 2015), circular correlation $corr(\cdot) = \boldsymbol{e}_j \circ \boldsymbol{r}$ (Nickel et al., 2016).

Subsequently, StarE and G-MPNN extend the applications of GNNs from KGs to HKGs. StarE (Galkin et al., 2020) requires the role information $r_{o_j}$ of entity $e_j$ and assumes a hyper-relational fact $r(e_1, \ldots, e_n)$ is composed by a base triplet $r(e_s, e_o)$ with a set of role-value pairs $\{(r_{o_j}, e_j)\}$ such that $r(e_s, e_o, \{(r_{o_j}, e_j)\})$. Then StarE takes base relation with the role-value pairs as a hyper-relation and performs the composition operator $\phi_r(\cdot)$ on the hyper-relation with base entity:

$$\boldsymbol{e}_s = act(sum(\{\boldsymbol{W}_{\lambda(r)} \phi_r(\boldsymbol{e}_o, \gamma(\boldsymbol{r}, \boldsymbol{h}_r))\}_{r(e_s, e_o, \{(r_{o_j}, e_j)\}) \in I(e_s)})), \ \boldsymbol{h}_r = \boldsymbol{W} sum(\{\phi_o(\boldsymbol{r}_{o_j}, \boldsymbol{e}_j)\}_j),$$

where $\boldsymbol{h}_r$ is the hidden representation aggregated from the role-value pairs, $\gamma(\cdot)$ is a concatenate operator to output the representation of hyper-relation. The update of $\boldsymbol{r}$ in StarE is similar to CompGCN. Instead of requiring role information, G-MPNN (Yadati, 2020) proposes to model positional information in GNNs:

$$\boldsymbol{e}_i = act([\boldsymbol{e}_i, agg(\{\boldsymbol{r}_{s, P(s)} * \prod_{j \in \{1, \ldots, n\}} \boldsymbol{p}_{e_j, s} * \boldsymbol{e}_j\}_{r(e_1, \ldots, e_n) \in I(e_i)})]), \tag{4}$$

where $s$ represents $r(e_1, \ldots, e_n) \in S$, $P(s) : s \to \{1, \ldots, n_p\}$ is a positional mapping ($n_p \leq |E|$), and $\boldsymbol{p}_{e_j, s}$ is the positional embedding vector of $e_j$ on fact $s$. To intuitively check the difference of message functions, we plot the framework of general MPPNs and GNNs for KGs/HKGs in Fig. 1.

## 3 MSEAHKG

In this section, we first propose a search space, especially the space of message functions, which enables the powerful GNNs to be searched for any given HKGs. Then, we formulate our search problem on the proposed space and solve it by leveraging an efficient search algorithm.

### 3.1 GNN SEARCH SPACE DESIGN FOR HKG EMBEDDING

As discussed in Sec. 1, designing the proper message function is more conducive to capturing relational patterns of given HKGs and pursuing high empirical performance. Therefore, the first task is to design an expressive search space so that message functions covering various relational patterns can be searched. However, the space of existing GNN searching methods (see Eq. 1 and Fig. 1) does not apply to HKGs and cannot cover the message functions of GNNs for KGs/HKGs (see Sec. 2.2). Thus, we focus on the search space design of message functions in this paper.

Recently, various message functions have been proposed for KGs/HKGs as presented in Sec. 2.2. From Fig. 1, we can observe that those message functions are mainly different in these two aspects: 1) the operators (e.g., $\boldsymbol{W}, \phi, \gamma$) for computing hidden representations, 2) the structure of message functions that decides how computational operators are connected. For the operator selection, existing works manually tune them on different data sets, such as $\phi$ in CompGCN, $\phi_o, \phi_r, \gamma$ in StarE, $agg(\cdot)$ in G-MPNN. Moreover, the structure of message functions for HKGs (StarE and G-MPNN) tends to be deeper and more complex than those for KGs (e.g., CompGCN). This is because the message function needs to process more information (e.g., more entities/roles) when facing the facts with higher arity. These observations motivate us to build space of operators and structures for message function search. Furthermore, we investigate more about the relationship between operators and relational patterns. Generally, the relational pattern can be represented as a certain correlation among $r(pemu(e_1, \ldots, e_n))$ (check more in Appx. B.2), where $pemu$ denotes the permutation. For example, $r$ is symmetry if $r(e_j, e_i)$ must be true when $r(e_i, e_j)$ is true. Therefore, the message function in the search space must be able to handle such correlation in the form of $r(pemu(e_1, \ldots, e_n))$. In the next, we first introduce the space of operators, and then discuss how they deal with $r(pemu(e_1, \ldots, e_n))$.

- **Positional Transformation Matrix $\boldsymbol{W}_{P(e)}$:** The position of entity in a fact can largely determine the plausibility of the fact. For example, `isCaptialOf(Beijing,China)` is true while `isCaptialOf(China,Beijing)` is false. G-MPNN utilizes the positional embedding $\boldsymbol{p}_{e,s}$ and $\boldsymbol{r}_{e,P(e)}$ to encode the position of entity $e$ and relation $r$ in different facts, which requires the model complexity $O(d|S|(|E| + N))$. However, the training data set is very sparse in KBs. Such over-parameterization may make the training insufficient. Instead, we adopt the way to transform one entity $e$ to $N$ possible positions. Let the positional mapping be $P(e) : e \to \{1, \ldots, N\}$, then the positional matrix is able to transform $\boldsymbol{e}$ to the permutation position in $pemu(\cdot)$ by $\boldsymbol{W}_{P(e)}\boldsymbol{e}$, where $\boldsymbol{W}_{P(e)}$ consumes $O(Nd^2)$ ($|S| \gg d$ in practice).

- **Concatenate Operator $\gamma(\cdot)$:** It mainly determines the concatenation way between embedding vectors. In this paper, we set $\mathcal{O}_\gamma = \{concat, mult, wsum\}$ (Galkin et al., 2020), where $wsum$ is the weighted sum. In general, $\gamma(\cdot)$ can concatenate embeddings after the positional transform matrix $\boldsymbol{W}_{P(e)}$, i.e., encoding $pemu(\boldsymbol{e}_1, \ldots, \boldsymbol{e}_n)$.

- **Role Embedding $\boldsymbol{r}_o$:** Unlike the positional information of entities encoded by $\boldsymbol{W}_{P(e)}$, the role embedding is utilized to model the semantic information of entities (Liu et al., 2021). For example, the roles in 2nd position of facts `playCharacerIn(actor:ZacharyQuinto,character:Spock,movie:StarTrek)` and `actorAwardIn(actor:ZacharyQuinto,award:BC-BSFC,movie:StarTrek)` are different though other entities and roles are same. Thus, the model should be able to capture the role of candidate entities after using $\boldsymbol{W}_{P(e)}$, e.g., `BC-BSFC` is unlike to be the 2nd entity of `playCharacerIn` since its role is `award` instead of `character`.

- **Composition Operator $\phi(\cdot)$:** Following CompGCN, we utilize composition operator $\phi(\cdot)$ to capture message between the node and edge embeddings before aggregation step. Note that $\phi$ actually encodes the interaction between $\boldsymbol{r}$ and $pemu(\boldsymbol{e}_1, \ldots, \boldsymbol{e}_n)$. While CompGCN and StarE empirically selects the most proper $\phi(\cdot)$, we include this operator into the space $\mathcal{X}$. Thus, MSeaHKG is able to automatically search for the most suitable $\phi(\cdot)$ in a more efficient way. We combine the settings of CompGCN and StarE to set $\mathcal{O}_\phi = \{sub, mult, corr, rotat\}$ ($rotat$ (Sun

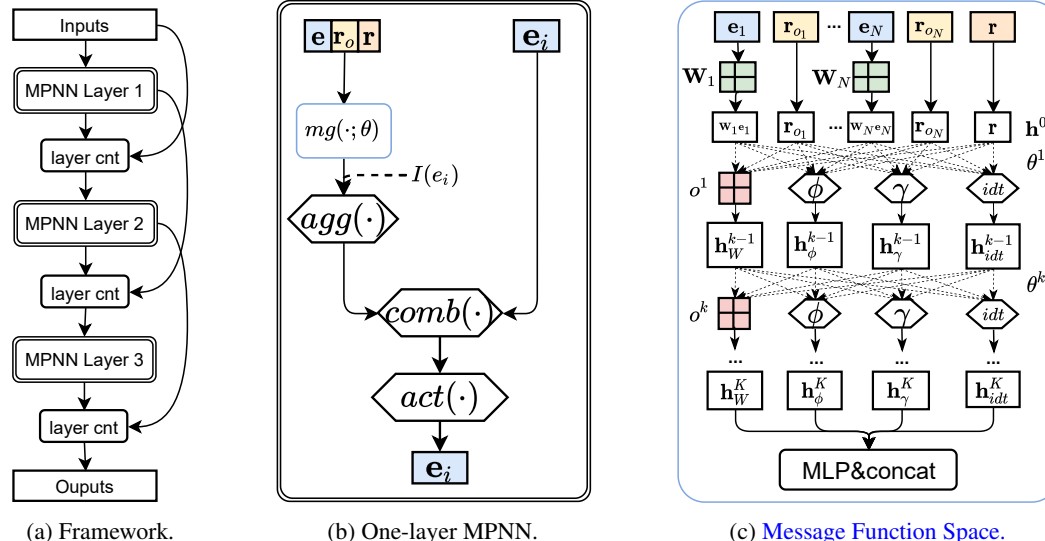

(a) Framework.          (b) One-layer MPNN.          (c) Message Function Space.

Figure 2: The framework of MSeaHKG. Fig. (b) represents concrete framework of MPNN layer in Fig. (a), and Fig. (c) is detailed formulation of $mg(\cdot; \boldsymbol{\theta})$ in Fig. (b). The operator $layer\ cnt$ enables the connectivity of different layers (Jiang & Balaprakash, 2020).

et al., 2019), see others in Sec 2.2). Besides, $\phi$ not only occurs between primary relations with entities, but also captures the correlation between roles with entities.

- **Others:** (1) The $idt(\boldsymbol{h}) = \boldsymbol{h}$ operation allows inputs to skip one layer in the message function; (2) Unlike $\boldsymbol{W}_{P(e)}$, the transform matrix $\boldsymbol{W}$ processes the hidden representations.

Among above components, we fix the role embedding and positional transform matrix $\boldsymbol{W}_{P(e)}$ in the message function (see Fig. 2 (c)) at the initial layer, which are specific designs to HKG problems. And we include others into the space of message function $\mathcal{O} = \{\boldsymbol{W}, \phi, \gamma, idt\}$ for searching. As shown in Fig. 2 (c), we denote the node $o_i^k$ as $i$-th operator of $\mathcal{O}$ in $k$-th layer and $\boldsymbol{h}_i^k$ be the hidden representation outputted by $o_i^k$. Then we have:

$$\{\boldsymbol{h}_i^0\}_{i=1}^{2n+1} = \{\boldsymbol{W}_{P(e_1)}\boldsymbol{e}_1, \ldots, \boldsymbol{W}_{P(e_n)}\boldsymbol{e}_n, \boldsymbol{r}_{o_1}, \ldots, \boldsymbol{r}_{o_n}, \boldsymbol{r}\}, \qquad (5)$$

$$\boldsymbol{h}_i^k = o_i^k\big(\{\theta_{ij}^k \boldsymbol{h}_j^{k-1}\}_{j=1}^{|\mathcal{O}|}\big), \text{for } k \in \{1, \ldots, K\} \text{ and } i \in \{1, \ldots, |\mathcal{O}|\}, \qquad (6)$$

where $\theta_{ij}^k \in \{0, 1\}$ controls the connection between $o_i^k$ with $o_j^{k-1}$. Note that $\{\boldsymbol{h}_i^0\}_{i=1}^{n+1} = \{\boldsymbol{W}_{P(e_1)}\boldsymbol{e}_1, \ldots, \boldsymbol{W}_{P(e_n)}\boldsymbol{e}_n, \boldsymbol{r}\}$ if role information is not available in the given HKG. From Fig. 2 (c), we can observe that the structure of message functions is controlled by $\{\theta_{ij}^k\}$.

To avoid manual operation selection, we also search for concrete operations of two operators $\phi$ and $\gamma$. Given the operator set $\mathcal{O}_\phi$ and $\mathcal{O}_\gamma$, let $\theta_i^{\phi_k}, \theta_i^{\gamma_k} \in \{0, 1\}$ records the selection $i$-th operation $o_i \in \mathcal{O}_\phi, \mathcal{O}_\gamma$ at $k$-layer respectively. Then, $\phi$ and $\gamma$ perform the computation in Eq. 6 could be $\phi^k(\boldsymbol{h}) = \sum \theta_i^{\phi_k} o_i(\boldsymbol{h})$ and $\gamma^k(\boldsymbol{h}) = \sum \theta_i^{\gamma_k} o_i(\boldsymbol{h})$. Note that $\sum_i \theta_i^{\phi_k} = 1$ and $\sum_i \theta_i^{\gamma_k} = 1$. Let $\boldsymbol{\theta}^{mg} = \{\theta_{ij}^k\} \cup \{\theta_i^{\phi_k}\} \cup \{\theta_i^{\gamma_k}\}$. The message function parameterized by $\boldsymbol{\theta}^{mg}$ is defined as:

$$mg\big(\boldsymbol{r}(\boldsymbol{e}_1, \ldots, \boldsymbol{e}_n); \boldsymbol{\theta}^{mg}\big) = MLP\&concat(\{\boldsymbol{h}_i^K\}_{i=1}^{|\mathcal{O}|}), \qquad (7)$$

where we simplify role embedding $\boldsymbol{r}_o$ for simplicity and $\boldsymbol{h}_i^K$ is outputted by the last layer of Eq. 6. Intuitively, existing GNNs for KGs/HKGs are contained in the MSeaHKG space (compare Fig. 1 and Fig. 2). Moreover, we include more discussions about the instantiations of other KG/HKGs models in the MSeaHKG space and the capability of handling relational patterns in Appx. B.2.

In addition to message function search, we also search for other operators (e.g., $agg, comb, act$) like existing GNN search methods as shown in Fig. 2 (a) and (b). Let $\boldsymbol{\Theta} = \{\boldsymbol{\theta}^{mg}, \boldsymbol{\theta}^{agg}, \ldots\}$ be parameter set for all operators selection in our MPNN framework. Then, an architecture can be represented as $X_{\boldsymbol{\Theta}} = \{mg(\cdot; \boldsymbol{\theta}^{mg}), agg(\cdot; \boldsymbol{\theta}^{agg}), \cdots\}$ (check other functions in Appx. C.1). And existing GNNs for KG/HKG embeddings actually can be represented by different instantiations of $\boldsymbol{\Theta}$. Overall, a GNN model $X_{\boldsymbol{\Theta}}$ encodes the given HKG $\mathcal{G}$ into embedding space $\boldsymbol{\omega} = \{\boldsymbol{E}, \boldsymbol{R}\}$, i.e., $\boldsymbol{\omega} = X_{\boldsymbol{\Theta}}(\mathcal{G})$.

## 3.2 Search Algorithm Design

In this subsection, we introduce how to select the GNN $X_{\Theta}$ that can achieve high performance on the given $\mathcal{G}$. First, we evaluate performance of $X_{\Theta}$ based on the HKG embedding $\boldsymbol{\omega} = \{\boldsymbol{E}, \boldsymbol{R}\}$ since $X_{\Theta}$ is utilized to encode $\mathcal{G}(E, R, S)$ into $\boldsymbol{\omega}$, i.e., $\boldsymbol{\omega} = X_{\Theta}(\mathcal{G})$. In the KG/HKG embedding, the scoring function $f(s; \boldsymbol{\omega})$ is to verify the plausibility of fact $s = r(e_1, \ldots, e_n)$ by decoding $\boldsymbol{\omega}$ (check more about scoring functions in Appx. C.2). Generally, the good embedding $\boldsymbol{\omega}$ can make $f(s; \boldsymbol{\omega})$ to distinguish true or false for a given fact $s$. Thus, we build evaluation of $X_{\Theta}$ on $f(s; \boldsymbol{\omega})$. Taking link prediction task (predict the missing entity in a fact, e.g., $r(?, e_2, \ldots, e_n)$) as example, let a scoring function $f(s; \boldsymbol{\omega})$ decode $\boldsymbol{\omega}$ and output a score matrix $\boldsymbol{p}^s \in [0, 1]^{|E|}$, where $\boldsymbol{p}^s_e$ is the probability score of $e \in E$ may be the ground truth of the missing entity. Then, we follow Dettmers et al. (2018) to construct the cross entropy loss $\ell(f(s; \boldsymbol{\omega})) = \sum_{e \in E} \boldsymbol{y}^s_e \log \boldsymbol{p}^s_e$, where $\boldsymbol{y}^s_e = 1$ if $e$ is the ground truth otherwise 0. Subsequently, we denote $\mathcal{L}(X_{\Theta}, \boldsymbol{\omega}; \mathcal{G}) = \sum_{s \in S} \ell(f(s; \boldsymbol{\omega}))$ to calculate the overall loss of a GNN model $X_{\Theta}$ with the HKG embedding $\boldsymbol{\omega}$ on $\mathcal{G}(E, R, S)$. Formally the GNN search problem for a given HKG $\mathcal{G}$ is formulated as:

$$\min_{\Theta, \boldsymbol{\omega}} \mathcal{L}(X_{\Theta}, \boldsymbol{\omega}; \mathcal{G}). \tag{8}$$

Solving Eq. 8 is a non-trivial task because $X_{\Theta}$ (e.g., $X_{\Theta} = \{agg : sum, act : relu, \ldots\}$) is from a large space. For example, only the structure space size $\{\theta^k_{ij}\}$ of message function reaches to $O(2^{K|\mathcal{O}|^2 + (2N+1)|\mathcal{O}|})$. And $X_{\Theta}$ is discrete, indicating the gradient-based optimization cannot be employed since $\nabla_{X_{\Theta}} \mathcal{L}(\cdot)$ does not exist. To enable an efficient search, we first relax the parameters of GNN model $\Theta$ from a discrete space into a continuous and probabilistic space $\bar{\Theta}$. More specifically, $\theta^k_{ij} \in \{0, 1\}$ restrictively controls the connectivity between $o^k_i$ with $o^{k-1}_j$, while $\bar{\theta}^k_{ij} \in [0, 1]$ is the probability that $o^k_i$ is connected with $o^{k-1}_j$. Then, let $X \sim p_{\bar{\Theta}}(X)$ represent a GNN model $X$ being sampled from the distribution $p_{\bar{\Theta}}(X)$. We reformulate the problem in Eq. 8 into:

$$\min_{\bar{\Theta}, \boldsymbol{\omega}} E_{X \sim p_{\bar{\Theta}}(X)}[\mathcal{L}(X, \boldsymbol{\omega}; \mathcal{G})], \tag{9}$$

where $E[\cdot]$ is the expectation. To compute the gradient w.r.t. $\bar{\Theta}$, we first utilize the reparameterization trick $X = g_{\bar{\Theta}}(U)$ (Maddison et al., 2016), where $U$ is sampled from a uniform distribution $p(U)$. Then the gradient w.r.t. $\bar{\Theta}$ and $\boldsymbol{\omega}$ is computed as (check full derivation in Appx. C.4):

$$\nabla_{\bar{\Theta}} E_{X \sim p_{\bar{\Theta}}(X)}[\mathcal{L}(X, \boldsymbol{\omega}; \mathcal{G})] = E_{U \sim p(U)}[\mathcal{L}'(g_{\bar{\Theta}}(U), \boldsymbol{\omega}; D) \nabla_{\bar{\Theta}} g_{\bar{\Theta}}(U)], \tag{10}$$

$$\nabla_{\boldsymbol{\omega}} E_{X \sim p_{\bar{\Theta}}(X)}[\mathcal{L}(X, \boldsymbol{\omega}; \mathcal{G})] = E_{X \sim p_{\bar{\Theta}}(X)}[\nabla_{\boldsymbol{\omega}} \mathcal{L}(X, \boldsymbol{\omega}; \mathcal{G})]. \tag{11}$$

Note that $\nabla_{\bar{\Theta}} g_{\bar{\Theta}}(U)$ can be computed if $g_{\bar{\Theta}}(U)$ is differentiable. Inspired by SNAS (Xie et al., 2018), we leverage Maddison et al. (2016); Jang et al. (2016) to instantiate $g_{\bar{\Theta}}(U)$ (see Appx. C.4).

## 4 Experiments

### 4.1 Experimental Setup

The experiments are implemented on top of PyTorch (Paszke et al., 2019) and performed on one single RTX 2080 Ti GPU. Appx. D.1.1 introduces the details of hyper-parameters.

**Data Sets.** The details of data sets are summarized into Tab. 7 in Appx. D.1.2. For experiments on HKGs (facts with mixed arities), we employ 2 benchmark data sets: (1) Wiki-People (Guan et al., 2019) is extracted from wiki-data, which mainly concerns the entities that belong to the human type. (2) JF17K (Zhang et al., 2018) is extracted from Freebase (Bollacker et al., 2008). For experiments on facts with fixed arities, we utilize several data sets with fixed arities and $n > 2$: WikiPeople-3, JF17k-3, WikiPeople-4, and JF17k-4. Note that GETD (Liu et al., 2020) filters out the 3-ary and 4-ary facts from WikiPeople and JF17K to construct WikiPeople-$n$ and JF17k-$n$, respectively.

**Tasks and Evaluation Metrics.** In this paper, we mainly compare HKG embedding models on the link and relation prediction task in the transductive setting. The link prediction task is to predict the missing entity in the given fact at $n$ possible positions, e.g., predicting the first missing entity $r(?, e_2, \cdots, e_n)$. The relation classification task needs to predict the missing relation in a fact when all entities are known, i.e., $?(e_1, \cdots, e_n)$. We employ Mean Reciprocal Ranking (MRR) (Voorhees, 1999) and Hits@$\{1, 3, 10\}$. All reports are in the "filtered" setting (see more in Appx. D.1.3).

Table 2: The model comparison of the link prediction task on HKGs. The results of NNs and multi-linear baselines are copied from Liu et al. (2020), those of Geo and S2S are copied from Di et al. (2021). And GNN baselines are re-implemented due to the task variance.

| type | model | WikiPeople | | | | JF17K | | | |
|---|---|---|---|---|---|---|---|---|---|
| | | MRR | Hit@1 | Hit@3 | Hit@10 | MRR | Hit@1 | Hit@3 | Hit@10 |
| **Geo** | RAE | 0.172 | 0.102 | 0.182 | 0.320 | 0.310 | 0.219 | 0.334 | 0.504 |
| **NNs** | NaLP | 0.338 | 0.272 | 0.364 | 0.466 | 0.366 | 0.290 | 0.391 | 0.516 |
| | HINGE | 0.333 | 0.259 | 0.361 | 0.477 | 0.473 | 0.397 | 0.490 | 0.618 |
| | NeuInfer | 0.350 | 0.282 | 0.381 | 0.467 | 0.517 | 0.436 | 0.553 | 0.675 |
| **Multi-linear** | HypE | 0.292 | 0.162 | 0.375 | 0.502 | 0.507 | 0.421 | 0.550 | 0.669 |
| | RAM | 0.380 | 0.279 | 0.445 | 0.539 | 0.539 | 0.463 | 0.573 | 0.690 |
| **GNNs** | StarE | 0.378 | 0.265 | 0.452 | 0.542 | 0.542 | 0.454 | 0.580 | 0.685 |
| | G-MPNN | 0.367 | 0.258 | 0.439 | 0.526 | 0.530 | 0.459 | 0.572 | 0.688 |
| **Search** | S2S | 0.372 | 0.277 | 0.439 | 0.533 | 0.528 | 0.457 | 0.570 | 0.690 |
| | MSeaHKG | **0.395** | **0.291** | **0.470** | **0.554** | **0.577** | **0.481** | **0.599** | **0.711** |

Table 3: The model comparison of the relation prediction task on HKGs. The results of NNs are copied from Guan et al. (2020), others are based on our implementations.

| type | model | WikiPeople | | | | JF17K | | | |
|---|---|---|---|---|---|---|---|---|---|
| | | MRR | Hit@1 | Hit@3 | Hit@10 | MRR | Hit@1 | Hit@3 | Hit@10 |
| **NNs** | NaLP | 0.735 | 0.595 | 0.852 | 0.938 | 0.825 | 0.762 | 0.873 | 0.927 |
| | NeuInfer | 0.765 | 0.686 | 0.877 | 0.900 | 0.861 | 0.832 | 0.885 | 0.910 |
| **GNNs** | StarE | 0.800 | 0.753 | 0.936 | 0.951 | 0.901 | 0.884 | 0.929 | 0.963 |
| | G-MPNN | 0.777 | 0.694 | 0.905 | 0.912 | 0.864 | 0.842 | 0.883 | 0.917 |
| **Search** | S2S | 0.813 | 0.744 | 0.928 | 0.960 | 0.912 | 0.877 | 0.932 | 0.951 |
| | MSeaHKG | **0.831** | **0.787** | **0.955** | **0.972** | **0.933** | **0.894** | **0.950** | **0.972** |

**Baselines.** Except for GNNs in Sec. 2.2, we present key functions of most adopted baselines in Tab. 5. For tasks on HKGs, we mainly compare the proposed method with advanced HKG embedding models: (1) Geometric model: RAE (Zhang et al., 2018), which is upgrade version of m-TransH (Wen et al., 2016) that extended from TransH (Wang et al., 2014); (2) GNN-based models: G-MPNN (Yadati, 2020) and StarE (Galkin et al., 2020) (note that we re-implement and tune them because G-MPNN is under inductive settings and StarE only tests the performance of main triplets in hyper-relational facts); (3) Other NN-based models: NaLP (Guan et al., 2019), HINGE (Rosso et al., 2020), and NeuInfer (Guan et al., 2020); (4) Multi-linear models: The final score of HypE (Fatemi et al., 2020) and RAM (Liu et al., 2021) is computed by multi-way inner product which is extended from bilinear KG embedding models (Yang et al., 2015). (5) Search method: S2S (Di et al., 2021) proposes a search space for tensor decomposition models and searches for sparse core tensors. It shares embeddings to jointly learn from facts with mixed arities to alleviate the issue of tensor modeling.

Except for the above methods on HKGs, we also include the tensor decomposition models (n-CP, n-TuckER, GETD) that work well on the scenario of facts with fixed arity. n-CP (Lacroix et al., 2018) leverages CANDECOMP/PARAFAC decomposition (Hitchcock, 1927), while n-TuckER (Balazevic et al., 2019) is based on Tucker decomposition (Tucker, 1966). GETD (Liu et al., 2020) proposes to reduce the model complexity by tensor ring decomposition.

## 4.2 Main Experimental Results

The main results on HKGs (i.e., WikiPeople and JF17K) have been summarized into Tab. 2 and Tab. 3. Compared with Geometric and classic NN-based methods (e.g., NaLP), GNNs methods (e.g., StarE) achieve outstanding performance, which demonstrates the power of GNNs on the graph tasks. And StarE generally is better than G-MPNN in GNNs methods because the inner product way in G-MPNN cannot handle several relational patterns as mentioned in Sec. 1. Besides, although the multi-linear method RAM utilizes the simple inner product as its scoring function, it carefully models the role semantic information and interaction patterns, thus achieving good performance. Another searching method S2S alleviates the extension issue of tensor decomposition models from the fixed to the mixed scenario. It is still slightly inferior compared with other state-of-the-art methods. Overall, all existing methods cannot consistently achieve the leading performance on different tasks and data sets. In this

Table 4: The model comparison of the link prediction task on facts with fixed arity. The results of tensor models are copied from Liu et al. (2020), others are copied from Di et al. (2021).

| type | model | WikiPeople-3 | | JF17K-3 | | WikiPeople-4 | | JF17K-4 | |
|------|-------|------|------|------|------|------|------|------|------|
| | | MRR | H@10 | MRR | H@10 | MRR | H@10 | MRR | H@10 |
| **Geo** | RAE | 0.239 | 0.379 | 0.505 | 0.644 | 0.150 | 0.273 | 0.707 | 0.835 |
| **NNs** | NaLP | 0.301 | 0.508 | 0.515 | 0.679 | 0.342 | 0.540 | 0.719 | 0.805 |
| | HINGE | 0.338 | 0.508 | 0.587 | 0.738 | 0.352 | 0.557 | 0.745 | 0.842 |
| | NeuInfer | 0.355 | 0.521 | 0.622 | 0.770 | 0.361 | 0.566 | 0.765 | 0.871 |
| **Tensor** | n-CP | 0.330 | 0.496 | 0.700 | 0.827 | 0.265 | 0.445 | 0.787 | 0.890 |
| | n-TuckER | 0.365 | 0.548 | 0.727 | 0.852 | 0.362 | 0.570 | 0.804 | 0.902 |
| | GETD | 0.373 | 0.558 | 0.732 | 0.856 | 0.386 | 0.596 | 0.810 | 0.913 |
| **Search** | S2S | 0.386 | 0.559 | 0.740 | 0.860 | 0.391 | 0.600 | 0.822 | 0.924 |
| | MSeaHKG | **0.405** | **0.583** | **0.757** | **0.892** | **0.412** | **0.628** | **0.834** | **0.940** |

paper, MSeaHKG pursues the high model performance by dynamically designing the most suitable message function for the given HKG and task. The searched message functions can capture data-level properties (see case study in Appx. D.2), thereby showing the leading performance. Especially, the search space of another search method S2S is based on the tensor modeling. As discussed in Sec. 1 and Appx. A, the MRHG could be a more natural way to represent HKGs, thereby MSeaHKG benefits from building a message function search space in GNNs under the MRHG modeling.

We show experiments on facts with fixed arity in Tab. 4. As discussed in Sec. 1, we can indeed observe that classic tensor decomposition models (n-CP, n-TuckER, GETD) perform better than Geometric and NN-based methods on the scenario of fixed arity. Then, S2S proposes to dynamically sparsify the core tensor of tensor decomposition models for the given data and further improve the performance of tensor models. Moreover, we note that MSeaHKG still performs better than S2S even in the scenario of fixed arity. That is because S2S simply assumes 3 relationships between entities and relations in the search space: positive, irrelevant, and negative. But the message function space in Sec. 3.1 could characterize more complex interactions between entities and relations, thereby achieving improvements.

### 4.3 MORE INSIGHTS VIA EMPIRICAL STUDY

Due to the space limitation, we include more experimental results in Appx. D to provide more insights and verify several claims in this paper. First, we demonstrate case studies in Appx. D.2 that the searched message functions are data-dependent and can adapt to the given data set. Second, we conduct ablation studies to analyze the components of the proposed method in Appx. D.3. Specifically, we show several variants of the proposed search space to demonstrate: (1) with a simple extension of message function, the GNN searching method cannot work well on HKGs (e.g., AutoGEL discussed in Sec. 2.2), (2) automatic operation selection can improve the performance built on the manual operation tuning, (3) the structure design of message functions is important to improve performance on HKGs. Then, we implement two more popular NAS algorithms (Liu et al., 2018; Akimoto et al., 2019) to compare the searching effectiveness and efficiency of one-shot NAS search algorithms in our scenarios. Third, we transfer MSeaHKG to more variety of graph-based tasks in Appx. D.4.

## 5 CONCLUSION

In this paper, we propose a new message function searching method for HKGs, named MSeaHKG. First, we present a novel search space of message functions in MPNNs, which enables both structure search and operation selection in message functions. With our expressive message function design, some classic KGs/HKGs models and existing message functions for HKGs could be instantiated as special cases of the proposed space. Then, we develop an efficient one-shot NAS algorithm to search the message function and other GNN components for the given HKG. The empirical study demonstrates that the searched message functions are data-dependent and can adapt to the data patterns of the given HKGs. Overall, MSeaHKG has shown its effectiveness and efficiency on benchmark data sets.

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

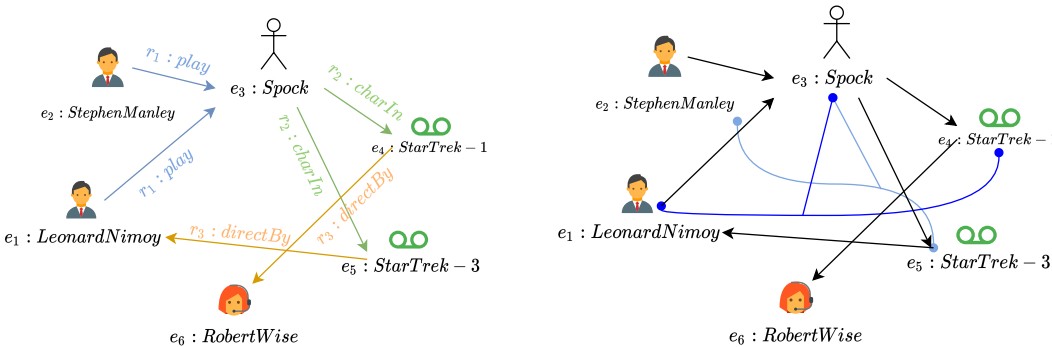

(a) KG with Multi-relational Graph Modeling.   (b) HKG with Multi-relational Hypergraph Modeling.

Figure 3: The illustration to KGs and HKGs. The binary facts are simplified in (b) to make the 3-ary facts look more prominent.

Richong Zhang, Junpeng Li, Jiajie Mei, and Yongyi Mao. Scalable instance reconstruction in knowledge bases via relatedness affiliated embedding. In WWW, pp. 1185–1194, 2018.

Yongqi Zhang, Quanming Yao, Wenyuan Dai, and Lei Chen. AutoSF: Searching scoring functions for knowledge graph embedding. In ICDE, pp. 433–444, 2020.

Ziwei Zhang, Xin Wang, and Wenwu Zhu. Automated machine learning on graphs: A survey. arXiv preprint arXiv:2103.00742, 2021.

Huan Zhao, Quanming Yao, and Weiwei Tu. Search to aggregate neighborhood for graph neural network. arXiv preprint arXiv:2104.06608, 2021.

Wang Zhili, Di Shimin, and Chen Lei. Autogel: An automated graph neural network with explicit link information. In NeurIPS, 2021.

Kaixiong Zhou, Qingquan Song, Xiao Huang, and Xia Hu. Auto-gnn: Neural architecture search of graph neural networks. arXiv preprint arXiv:1909.03184, 2019.

## A  DISCUSSION ON THE MODELING OF HKG

Given a set of facts with the fixed arity $\{r(e_1, \ldots, e_n)\}$, tensor models (e.g., CP (Lacroix et al., 2018) and TuckER (Balazevic et al., 2019)) use a $(n+1)$-dimensional tensor $\boldsymbol{G}^{n+1} \in \{0,1\}^{|R| \times |E| \times \cdots \times |E|}$ to represent facts, where $\boldsymbol{G}^{n+1}_{r,1,\ldots,n} = 1$ represents the fact $r(e_1, \ldots, e_n)$ existed otherwise $\boldsymbol{G}^{n+1}_{r,1,\ldots,n} = 0$. Then, tensor models leverage the tensor decomposition techniques to decompose $\boldsymbol{G}^{n+1}$ into embeddings $\boldsymbol{R}$ and $\boldsymbol{E}$ with a core tensor $\boldsymbol{\mathcal{Z}}^{n+1} \in \mathbb{R}^{d \times \cdots \times d}$:

$$\boldsymbol{G}^{n+1} = \boldsymbol{\mathcal{Z}}^{n+1} \times_1 \boldsymbol{R} \times_2 \boldsymbol{E} \times_3 \cdots \times_{n+1} \boldsymbol{E}.$$

Obviously, a tensor $\boldsymbol{G}^{n+1}$ can only model facts with the fixed arity $n$. Thus, we have to build $\{\boldsymbol{G}^2, \ldots, \boldsymbol{G}^{N+1}\}$ to model the facts with mixed arities $S = \{r(e_1, \ldots, e_n) : n \in \{2, \ldots, N\}\}$ and decompose them into $N-1$ sets of entity and relation embeddings, such as $\{\boldsymbol{E}^n, \boldsymbol{R}^n\}_{n=2}^N$. But KBs are known to have the data sparse issue, thus learning multiple set of embeddings could lead to severe problems.

As mentioned in Sec. 1, multi-relational hypergraphs (MRHGs) $\mathcal{G}(E, R, S)$ could be a more natural way to represent HKGs. We illustrate examples of KGs and HKGs in Fig. 3 (a) and (b), respectively. Given a fact with high arity (e.g., playCharacterIn(e₁,e₃,e₅)), MRHG can first store the correspondence between multiple entities by a hyperedge $(e_1, e_3, e_5)$, then label this hyperedge with the relation playCharacterIn. Moreover, Fig. 3 (b) demonstrates that the facts with mixed arities ($n \in \{2, 3\}$ in this example) can be represented by a MRHG. Thus, MRHG allows us to directly encode a given HKG into entity and relation embeddings, where every embedding vector is jointly learned from low and high arities. However, tensor modeling leads the model to learn multiple sets of embeddings.

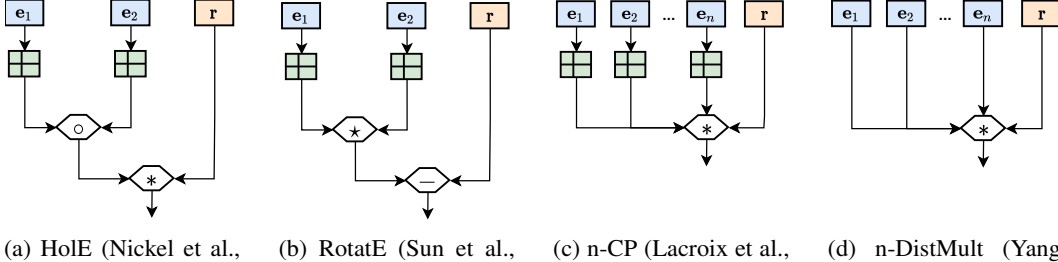

(a) HolE (Nickel et al., 2016).

(b) RotatE (Sun et al., 2019).

(c) n-CP (Lacroix et al., 2018).

(d) n-DistMult (Yang et al., 2015).

Figure 4: Several instantiation cases of MSeaHKG message function space. $\circ$ denotes circular correlation $corr(\cdot)$ (Nickel et al., 2016), $*$ denotes inner product $mult(\cdot)$ (Yang et al., 2015), $-$ denotes the substraction $sub(\cdot)$ (Bordes et al., 2013), $\star$ is from RotatE (Sun et al., 2018).

Besides, HKGs generally contain more complete information compared with KGs. For example, it is hard to answer "who plays Spock in StarTrek-3?" with the KG example (Fig. 3 (a)). That is because KGs suffer from the information loss of correspondence among multiple entities. Generally, the constructing procedure of KG, named Star-to-Clique Conversion, has been verified to be irreversible and caused the information loss in the facts with high arities (Wen et al., 2016). Therefore, we believe that HKG could be a potential solution to solve complex Q&A scenarios.

# B  DISCUSSION ON DATA-LEVEL PROPERTIES OF KGS/HKGS AND MESSAGE FUNCTION DESIGN

## B.1  DATA-LEVEL PROPERTIES IN KGS: RELATIONAL PATTERNS

In the past decades, the research community has found that the relations of KGs exhibit different patterns. For example, we can infer that `neighborOf(location2,location1)` must be true as long as `neighborOf(location1,location2)` is true, like `neighborOf(ShenZhen,HongKong)`. But for other relations (e.g., `largerThan`), we know `largerThan(value2,value1)` must be false if `largerThan(value1,value2)` is true. Many relational patterns in KGs have been found and discussed (Kazemi & Poole, 2018; Rossi et al., 2021), such as symmetry, anti-symmetry, asymmetry, inversion, composition, hierarchy, intersection. However, various KGs usually contain different relational patterns with different proportions of facts. To handle diverse relational patterns on different KGs (i.e., data-level properties), kinds of scoring functions have been proposed to cover as many relational patterns as possible, i.e., pursuing expressiveness. Given the learned embeddings $\boldsymbol{\omega} = \{\boldsymbol{E}, \boldsymbol{R}\}$, the scoring function $f(\cdot)$ is proposed to evaluate whether the fact is true or false (check Appx. C.2 and Tab. 5 for more details about scoring function). For the sake of understanding, we here present the requirements of several relational patterns on scoring functions:

- Symmetry: $r$ is symmetry if $r(e_j, e_i)$ must be true when $r(e_i, e_j)$ is true, i.e., $f(\boldsymbol{r}(\boldsymbol{e}_i, \boldsymbol{e}_j)) = f(\boldsymbol{r}(\boldsymbol{e}_j, \boldsymbol{e}_i))$

- Anti-symmetry: $r$ is anti-symmetric if $r(e_j, e_i)$ must be false when $r(e_i, e_j)$ is true, i.e., $f(\boldsymbol{r}(\boldsymbol{e}_i, \boldsymbol{e}_j)) = -f(\boldsymbol{r}(\boldsymbol{e}_j, \boldsymbol{e}_i))$.

- Asymmetry: The plausibility of $r(e_j, e_i)$ is unknown when $r(e_i, e_j)$ is true, $f(\boldsymbol{r}(\boldsymbol{e}_i, \boldsymbol{e}_j)) \neq f(\boldsymbol{r}(\boldsymbol{e}_j, \boldsymbol{e}_i))$.

- Inverse: $r_1$ and $r_2$ are inverse relations if $r_1(e_i, e_j)$ is true and $r_2(e_j, e_i)$ is true. Then, $f(\boldsymbol{r}_1(\boldsymbol{e}_i, \boldsymbol{e}_j)) = f(\boldsymbol{r}_2(\boldsymbol{e}_j, \boldsymbol{e}_i))$

As mentioned in Sec. 1, the message function of G-MPNN may not be able to infer the non-symmetric relations. Suppose that $s_1 = r(e_1, e_2)$ is true and $r$ is one of any non-symmetric relations (e.g., anti-symmetry $s_2 = r(e_2, e_1)$ must be false). Let $\boldsymbol{h}_i$ be the entity feature and $\boldsymbol{h}_r$ be the relation feature. According to the message function of G-MPNN in Eq. 4, we can learn the hidden embeddings of $\boldsymbol{m}_i$: $\boldsymbol{m}_1 = \boldsymbol{m}_2 = \boldsymbol{h}_r * \boldsymbol{h}_1 * \boldsymbol{p}_{e_1,s} * \boldsymbol{h}_2 * \boldsymbol{p}_{e_2,s}$ Then $\boldsymbol{e}_i$ are updated with $\boldsymbol{m}_i$. With the scoring function

Table 5: Scoring Functions for Facts with High-order Arities.

| Type | Model | Scoring Function Design $f(s; \boldsymbol{\omega})$ | Notes |
|---|---|---|---|
| **Geo** | m-TransH | $\left\| \sum_{i=1}^{n} a_j (\boldsymbol{e}_i - \boldsymbol{v}_r^\top \boldsymbol{e}_i \boldsymbol{v}_r) + \boldsymbol{r} \right\|^2$ | $\boldsymbol{v}_r$ is relation dependent. $FC(\cdot)$ is fully connected layer. |
| | RAE | $\left\| \sum_{i=1}^{n} a_j (\boldsymbol{e}_i - \boldsymbol{v}_r^\top \boldsymbol{e}_i \boldsymbol{v}_r) + \boldsymbol{r} \right\|^2 + \sum_{ij} FC([\boldsymbol{e}_i, \boldsymbol{e}_j])$ | |
| **Tensor** | n-CP | $\boldsymbol{r} * \boldsymbol{e}_1^{(1)} * \boldsymbol{e}_2^{(2)} * \cdots * \boldsymbol{e}_n^{(n)}$ | $\boldsymbol{e}_i^{(j)}$ is for $e_i$ in $j$-th position. $\boldsymbol{\mathcal{Z}} \in \mathbb{R}^{d \times \cdots \times d}$, $\times_i$ is tensor product in $i$-th mode. $TR(\cdot)$ is tensor ring decomposition $\boldsymbol{\mathcal{W}}_i$ is a 3-order tensor. |
| | n-TuckER | $\boldsymbol{\mathcal{Z}} \times_1 \boldsymbol{r} \times_2 \boldsymbol{e}_1 \times_3 \cdots \times_{n+1} \boldsymbol{e}_n$ | |
| | GETD | $TR(\boldsymbol{\mathcal{W}}_1, \ldots, \boldsymbol{\mathcal{W}}_k) \times_1 \boldsymbol{e}_1 \times_2 \boldsymbol{e}_2 \times_3 \cdots \times_{n+1} \boldsymbol{e}_n$ | |
| **GNNs** | G-MPNN | $\boldsymbol{r}_{s,P(s)} * \prod_{e_j \in Seen(s)} \boldsymbol{p}_{e_j,s} * \boldsymbol{e}_j$ | $Seen(s)$ returns the seen entities $e_j$ in $s$ |
| **Multi-linear** | HypE | $\boldsymbol{r} * Conv(\boldsymbol{e}_1) * Conv(\boldsymbol{e}_2) * \cdots * Conv(\boldsymbol{e}_n)$ | $Conv(\cdot)$ is the convolutional filter. $\boldsymbol{r}_i$ is embedding for $i$-th role of $r$, $\bar{\boldsymbol{e}}_j \in \mathbb{R}^{m \times d}$ is embedding matrix of $e$ |
| | RAM | $\sum_{i=1}^{n} \boldsymbol{r}_i * \boldsymbol{p}_i^r(1,:)\bar{\boldsymbol{e}}_1 * \cdots * \boldsymbol{p}_i^r(n,:)\bar{\boldsymbol{e}}_n$ | |
| **Search** | S2S | $\sum_{j_r, j_1, \ldots, j_s} \boldsymbol{\mathcal{Z}}_i^n \times_1 \boldsymbol{r}^{j_r} \times_2 \boldsymbol{e}_1^{j_1} \times_3 \cdots \times_{n+1} \boldsymbol{e}_n^{j_n}$ | $\boldsymbol{\mathcal{Z}}_i^n$ is the searched sparse core tensor, $i$ denotes $(j_r, j_1, \ldots, j_s)$. |

of G-MPNN (multiple inner product), we can compute $f(\boldsymbol{r}(\boldsymbol{e}_1, \boldsymbol{e}_2)) = \boldsymbol{h}_r * \boldsymbol{p}_{e_1,s} * \boldsymbol{e}_1 * \boldsymbol{p}_{e_2,s} * \boldsymbol{e}_2$. Because $e_1, e_2$ are seen in $s_1$, G-MPNN will assign the positional embeddings $e_1, e_2$ in $s_1$ when computing the score of $s_2$, thereby leading $f(\boldsymbol{r}(\boldsymbol{e}_1, \boldsymbol{e}_2)) = f(\boldsymbol{r}(\boldsymbol{e}_2, \boldsymbol{e}_1))$. Thus, we argue that the fixed message function is not flexible enough to capture the relational patterns of the given HKGs, especially its performance may not be good if some uncovered relational patterns exist in the given data set.

## B.2 DISCUSSION ON THE CAPABILITY OF MESSAGE FUNCTIONS TO COVER RELATIONAL PATTERNS ON HKGS

As mentioned in Appx. B.1, pursuing expressiveness is one way to cover relational patterns on KGs. Such potential solution is also discussed in Sec. 1. However, Meilicke et al. (2018); Rossi et al. (2021) report that being expressive does not mean achieving good performance even the model can cover those relational patterns. Thus, AutoSF (Zhang et al., 2020) proposes to search bilinear-based scoring functions for KGs and consistently achieve good performance. Inspired by literature, MSeaHKG proposes to search proper message functions for the pursuit of high model performance on any given HKGs.

Generally, the relational patterns on HKGs have not been explored before. Inspired by literature on KGs, we roughly summarize a high level representations of relational patterns in the scenario of high arity as:

$$f(r_1(pemu(e_1, \ldots, e_n)); \boldsymbol{\omega}) ? f(r_2(pemu(e_1, \ldots, e_n)); \boldsymbol{\omega}),$$

where $r_1, r_2$ may be same relation or different, $pemu(\cdot)$ represents the permutation of $n$ entities $\{e_i\}_{i=1}^{n}$, and $? \in \{=, >, \neq, \neg\}$. For example, $f(r_1(pemu(e_1, \ldots, e_n)); \boldsymbol{\omega}) = f(r_1(pemu(e_1, \ldots, e_n)); \boldsymbol{\omega})$ indicates $r_1$ is a symmetry. In this paper, since our scoring function design is a linear transformation (see Appx. C.2), the design of message function in MPNNs (i.e., $\boldsymbol{\omega} = X_{\boldsymbol{\Theta}}(\mathcal{G})$) will be the crucial component to capture relational patterns. To capture the permutation information $pemu(\cdot)$, we first utilize $\boldsymbol{W}_{P(e)}$ to encode the positional information, which transforms entity embeddings into corresponding positions. Then, the operator $\gamma(\cdot)$ concatenates the entity embeddings after encoding positional information. Lastly, the operator $\phi(\cdot)$ computes the interaction between $\boldsymbol{r}$ with $\boldsymbol{e}_i$. In other words, $\boldsymbol{W}_{P(e)}$ and $\gamma(\cdot)$ are employed to represent $pemu(\cdot)$, and $\phi(\cdot)$ is utilized to represent the correlation between $r$ and $pemu(\cdot)$. Within such design, most classic scoring function designs (including Geometric models (Bordes et al., 2013; Nickel et al., 2016; Sun et al., 2019), Bilinear models (Yang et al., 2015) and Tensor models (Lacroix et al., 2018)) and GNNs

for KGs/HKGs can be instantiated as special cases in the MSeaHKG space. We have demonstrated several examples of classic scoring function designs in Fig. 4. Note that the expressiveness of these models has been fully investigated in literature. Every model can cover one or multiple relational patterns, which can make our message function space expressive enough to handle most HKGs. In other words, the search space of MSeaHKG has the potential to return a GNN model that can adapt to the relational patterns of the given HKG.

## C  SUPPLEMENTARY OF SEARCH ALGORITHM

### C.1  THE PARAMETERIZATION OF OTHER GNN COMPONENTS

Generally, the relaxation of other GNN components (e.g., aggregation and activation) is consistent with $\phi^k(h) = \sum \theta_i^{\phi_k} o_i(h)$ and $\gamma^k(h) = \sum \theta_i^{\gamma_k} o_i(h)$ as presented in Sec. 3.1. Here we illustrate more details.

- Let $\mathcal{O}^{agg} = \{sum, mean, max\}$ be the set of candidate aggregation functions and $mg(\boldsymbol{r}(\boldsymbol{e}_1, \ldots, \boldsymbol{e}_n); \boldsymbol{\theta}^{mg})$ be the output from Eq. 7. Then, the aggregation function $agg(\cdot; \boldsymbol{\theta}^{agg})$ parameterized by $\boldsymbol{\theta}^{agg}$ can be defined as:

$$\boldsymbol{m}_i = agg(\{mg(\boldsymbol{r}(\boldsymbol{e}_1, \ldots, \boldsymbol{e}_n); \boldsymbol{\theta}^{mg})\}_{r(e_1, \ldots, e_n) \in I(e_i)}; \boldsymbol{\theta}^{agg})$$
$$= \sum_{o_j \in \mathcal{O}^{agg}} \theta_j^{agg} \cdot o_j(\{mg(\boldsymbol{r}(\boldsymbol{e}_1, \ldots, \boldsymbol{e}_n); \boldsymbol{\theta}^{mg})\}_{r(e_1, \ldots, e_n) \in I(e_i)}).$$

- Let $\mathcal{O}^{act} = \{identity, sigmoid, tanh\}$ be the set of candidate activation functions. Then, the activation function $act(\cdot; \boldsymbol{\theta}^{act})$ parameterized by $\boldsymbol{\theta}^{act}$ can be defined as:

$$\boldsymbol{e}_i = act(comb(\boldsymbol{e}_i, \boldsymbol{m}_i); \boldsymbol{\theta}^{act}) = \sum_{o_j \in \mathcal{O}^{act}} \theta_j^{act} \cdot o_j(comb(\boldsymbol{e}_i, \boldsymbol{m}_i))$$

Then, as presented in Sec. 3.2, we can relax $\boldsymbol{\theta}^{agg}$ and $\boldsymbol{\theta}^{act}$ into continuous space $\bar{\boldsymbol{\theta}}^{agg}$ and $\bar{\boldsymbol{\theta}}^{act}$ by leveraging the Gumbel-Softmax technique.

### C.2  SCORING FUNCTION FORMULATION

As mentioned in Sec. 3.2, the GNN model $X$ embeds the HKG $\mathcal{G}(E, R, S)$ into the low-dimensional vector space $\boldsymbol{\omega} = \{\boldsymbol{E}, \boldsymbol{R}\}$, i.e., $\boldsymbol{\omega} = X(\mathcal{G})$. Besides, it is also important to design a scoring function $f(s; \boldsymbol{\omega})$ to interprets the plausibility of the fact $s$ based on the learned $\boldsymbol{\omega}$ (Rossi et al., 2021; Zhang et al., 2020; Shimin et al., 2021). Many promising scoring functions have been proposed in the past decades, including geometric models (e.g., TransE (Bordes et al., 2013) and RotatE (Sun et al., 2018)), neural network models (e.g., ConvE (Dettmers et al., 2018)), tensor decomposition models (e.g., CP (Lacroix et al., 2018), TuckER (Balazevic et al., 2019)). Inspired by the success of literature, we regard the scoring function design as a component of implementations. In this paper, we focus on those scoring functions for facts with high arity ($n > 2$) and summarize the popular ones in Tab. 5. Note that StarE (Galkin et al., 2020) employs the Transformer (Vaswani et al., 2017) as its scoring function, thus it is not included in Tab. 5.

In principle, MSeaHKG can implement most of existing scoring functions as its decoder. But we argue that the power of the searched message function could well model the interaction between entities and relations in the encoding step. Within a powerful encoder, it may be unnecessary to introduce a complex scoring function like CompGCN (Vashishth et al., 2020) and StarE (Galkin et al., 2020) to decode embeddings. In this paper, we simply concatenate the embeddings of known entities and relations in a fact and feed it into the linear transformation with a softmax operator.

### C.3  LOSS FUNCTION W.R.T. RELATION PREDICTION

Note that the loss function $\mathcal{L}(X, \boldsymbol{\omega}; \mathcal{G})$ introduced in Sec. 3.2 is under the scenario of link prediction task. In the relation classification task, the probability score is formed to $\boldsymbol{p}^s \in [0, 1]^{|R|}$, i.e., predicting

the ground truth $r \in R$ when the relation is missing in $?(e_1, \ldots, e_2)$. The loss function is defined as:

$$\mathcal{L}(X, \boldsymbol{\omega}; \mathcal{G}) = \sum_{s \in S} \sum_{r \in R} \boldsymbol{y}_r^s \log \boldsymbol{p}_r^s,$$

where $\boldsymbol{y}_r^s = 1$ if $r$ is the ground truth relation in $s$.

## C.4 FULL DERIVATION INVOLVED IN SEC. 3.2

First, we present the full derivation of Eq. 10 and Eq. 11, and their approximation based on Monte-Carlo sampling:

$$\nabla_{\bar{\boldsymbol{\Theta}}} E_{X \sim p_{\bar{\boldsymbol{\Theta}}}(X)}[\mathcal{L}(X, \boldsymbol{\omega}; \mathcal{G})] = \nabla_{\bar{\boldsymbol{\Theta}}} E_{U \sim p(U)}[\mathcal{L}(g_{\bar{\boldsymbol{\Theta}}}(U), \boldsymbol{\omega}; \mathcal{G})] = \nabla_{\bar{\boldsymbol{\Theta}}} \int p(U) \mathcal{L}(g_{\bar{\boldsymbol{\Theta}}}(U), \boldsymbol{\omega}; \mathcal{G}) dU$$

$$= \int p(U) \nabla_{\bar{\boldsymbol{\Theta}}} \mathcal{L}(g_{\bar{\boldsymbol{\Theta}}}(U), \boldsymbol{\omega}; \mathcal{G}) dU = E_{U \sim p(U)}[\nabla_{\bar{\boldsymbol{\Theta}}} \mathcal{L}(g_{\bar{\boldsymbol{\Theta}}}(U), \boldsymbol{\omega}; \mathcal{G})]$$

$$= E_{U \sim p(U)}[\mathcal{L}'(g_{\bar{\boldsymbol{\Theta}}}(U), \boldsymbol{\omega}; \mathcal{G}) \nabla_{\bar{\boldsymbol{\Theta}}} g_{\bar{\boldsymbol{\Theta}}}(U)],$$

$$\nabla_{\boldsymbol{\omega}} E_{X \sim p_{\bar{\boldsymbol{\Theta}}}(X)}[\mathcal{L}(X, \boldsymbol{\omega}; \mathcal{G})] = \nabla_{\boldsymbol{\omega}} \int p_{\bar{\boldsymbol{\Theta}}}(X) \mathcal{L}(X, \boldsymbol{\omega}; \mathcal{G}) dX = \int p_{\bar{\boldsymbol{\Theta}}}(X) \nabla_{\boldsymbol{\omega}} \mathcal{L}(X, \boldsymbol{\omega}; \mathcal{G}) dX$$

$$= E_{X \sim p_{\bar{\boldsymbol{\Theta}}}(X)}[\nabla_{\boldsymbol{\omega}} \mathcal{L}(X, \boldsymbol{\omega}; \mathcal{G})].$$

Second, we build the reparameterization trick $X = g_{\bar{\boldsymbol{\Theta}}}(U)$ (mentioned in Sec. 3.2) based on Gumbel-Softmax (Jang et al., 2016) or Concrete distribution (Maddison et al., 2016). For simplicity, we simplify $\bar{\boldsymbol{\Theta}}$ to the parameter $\bar{\boldsymbol{\theta}}$ for a specific operator space $\mathcal{O}$:

$$X_o = g_{\bar{\boldsymbol{\theta}}}(U) = \frac{\exp((\log \bar{\boldsymbol{\theta}}_o - \log(-\log(U_o)))/\tau)}{\sum_{o' \in \mathcal{O}} \exp((\log \bar{\boldsymbol{\theta}}_{o'} - \log(-\log(U_{o'})))/\tau)}, \tag{12}$$

where $\tau$ is the temperature of softmax, and $U_o \sim Uniform(0, 1)$. It has been proven that $p(\lim_{\tau \to 0} X_o = 1) = \bar{\boldsymbol{\theta}}_o / \sum_{o' \in \mathcal{O}} \bar{\boldsymbol{\theta}}_{o'}$ making the stochastic differentiable relaxation unbiased once converged (Xie et al., 2018). And the details of $\nabla_{\bar{\boldsymbol{\Theta}}} g_{\bar{\boldsymbol{\Theta}}}(U)$ can refer to Xie et al. (2018).

## D MORE EXPERIMENTS

### D.1 EXPERIMENTAL SETUP

#### D.1.1 HYPER-PARAMETER SETTINGS

We have summarized the hyper-parameters of this paper in Tab. 6. The hyper-parameter set includes Adam optimizer (Kingma & Ba, 2014), learning rate $\in \{0.1, 0.01, 0.001, 0.0001, 0.00001\}$, # MPNN layers $\in \{1, 2, 3, 4\}$ (see left part of Fig. 2), # layers in message functions $K \in \{1, 2, 3, 4, 5\}$ (see right part of Fig. 2), batch size $\in \{64, 128, 256, 512\}$, embedding dimension $d \in \{64, 128, 256, 512\}$, dropout ratio $\in \{0, 0.05, 0.1, 0.15, \ldots, 0.5\}$, label smoothing ratio $\tau \in \{0, 0.1, 0.2, \ldots, 0.9\}$. Note that the label smoothing ratio $\tau$ is employed to relax the one-hot label vector $\boldsymbol{y}$. In practical, we set $\boldsymbol{y}_i = 1 - \tau$ for the ground truth entity/relation, while $\boldsymbol{y}_i = \tau/|E|-1$ for link prediction and $\boldsymbol{y}_i = \tau/|R|-1$ for relation prediction. That is because there are usually a large candidate space for relations and entities, while Using one-hot vector is quite restrictive. All hyper-parameters are tuned with the help of `optuna.samplers.TPESampler` (Bergstra et al., 2013; Falkner et al., 2018)[1].

#### D.1.2 DATA STATISTICS

Here we present the statistic summary of the benchmark data sets in this paper. And the links for download them.

---

[1] `https://optuna.readthedocs.io/en/stable/reference/generated/optuna.samplers.TPESampler.html`

Table 6: List of hyper-parameters in main experiments. W and J are abbreviations of WikiPeople and JF17K, respectively.

| Hyperparameters | Link Prediction | | | | | | Relation Prediction | |
|---|---|---|---|---|---|---|---|---|
| | WikiPeople | JF17K | W-3 | J-3 | W-4 | J-4 | WikiPeople | JF17K |
| Learning rate | 0.0001 | 0.001 | 0.0001 | 0.001 | 0.0001 | 0.0001 | 0.0001 | 0.001 |
| # MPNN layers | 2 | 2 | 1 | 1 | 1 | 1 | 2 | 2 |
| # Layers in $mg(\cdot)$ $K$ | 4 | 4 | 3 | 2 | 2 | 3 | 4 | 4 |
| Batch size | 256 | 128 | 128 | 128 | 256 | 128 | 256 | 128 |
| Embedding dim $d$ | 256 | 256 | 128 | 128 | 256 | 128 | 128 | 128 |
| Dropout ratio | 0.15 | 0.2 | 0.1 | 0.1 | 0.05 | 0.15 | 0.2 | 0.15 |
| Label smoothing ratio $\tau$ | 0.3 | 0.8 | 0.7 | 0.5 | 0.8 | 0.7 | 0.1 | 0.1 |

Table 7: The statistical summary on data sets.

| type | data set | # all facts | # facts ($n > 2$) | max arity $N$ | # ent | # rel | train | valid | test |
|---|---|---|---|---|---|---|---|---|---|
| Mixed | JF17K | 100,947 | 46,320 | 6 | 28,645 | 322 | 76,379 | - | 24,568 |
| | WikiPeople | 382,229 | 44,315 | 9 | 47,765 | 707 | 305,725 | 38,223 | 38,281 |
| Fixed | WN18RR | 93,003 | 0 | 2 | 40,943 | 11 | 86,835 | 3,034 | 3,134 |
| | FB15k237 | 310,116 | 0 | 2 | 14,541 | 237 | 272,115 | 17,535 | 20,466 |
| | JF17K-3 | 34,544 | 34,544 | 3 | 11,541 | 104 | 27,635 | 3,454 | 3,455 |
| | JF17K-4 | 9,509 | 9,509 | 4 | 6,536 | 23 | 7,607 | 951 | 951 |
| | WikiPeople-3 | 25,820 | 25,820 | 3 | 12,270 | 66 | 20,656 | 2,582 | 2,582 |
| | WikiPeople-4 | 15,188 | 15,188 | 4 | 9,528 | 50 | 12,150 | 1,519 | 1,519 |

### D.1.3 EVALUATION MEASUREMENT

Let $\text{rank}_{i,n}$ denote the rank of ground truth entity at position $n$ of $i$-th fact $s = (r, e_1, \ldots, e_N)$, defined as:

$$\text{rank}_{i,n} = |\{e' \in E \setminus \{e_n\} : f(s'; \boldsymbol{\omega}) > f(s; \boldsymbol{\omega})\}| + 1,$$

where $s' = (r, e_1, \ldots, e_{n-1}, e', \ldots, e_N)$. As in (Bordes et al., 2013; Wang et al., 2014), we report performance under the "filtered" setting, i.e., evaluating the rank of test fact after removing all corrupted facts that exist in the train, valid, and test data set. That is because true facts in data set should be not considered as faults when evaluating a test fact. Correspondingly, $\text{rank}_{i,n}$ under filtered setting is formed as:

$$\text{rank}_{i,n} = |\{e' \in E \setminus \{e_n\} : f(s'; \boldsymbol{\omega}) > f(s; \boldsymbol{\omega}) \cap s' \notin S\}| + 1.$$

Then we adopt the classical metrics (Bordes et al., 2013; Wang et al., 2014):

- Mean Reciprocal Ranking (MRR): $1/N|S| \sum_{i=1}^{|S|} \sum_{n=1}^{N} 1/\text{rank}_{i,n}$;

- Hit@1, Hit@3, and Hit@10, where Hit@k is given by $1/|S| \sum_{i=1}^{|S|} \sum_{n=1}^{N} \mathbb{I}(\text{rank}_{i,n} \leq k)$ and $\mathbb{I}(\cdot)$ is the indicator function.

Note that the higher MRR and Hit@k values mean better embedding quality.

### D.2 CASE STUDY

The relational patterns on KGs (see Appx. B) have been fully investigated (Kazemi & Poole, 2018; Rossi et al., 2021) in existing works. Thus, we first show the case studies on two benchmark KGs WN18RR (Dettmers et al., 2018) and FB15k237 (Toutanova & Chen, 2015) (see Tab. 7 for data statistics and Tab. 8 for experimental results), which have removed the duplicate and inverse relations of WN18 and FB15k (Bordes et al., 2013; Dettmers et al., 2018). As presented in Rossi et al. (2021), the irreflexive, anti-symmetric, and symmetric relations account for a major proportion of these two data sets, especially facts with the symmetric relations reach 37% in WN18RR. We found that the message function searched on WN18RR (Fig. 5 (a)) is the exact RotatE model (Sun et al., 2019), which has been proven to cover symmetric relations. As for the message function searched on FB15k237 (Fig. 5 (b)), it first feeds the relation embedding into a transformation matrix $\boldsymbol{W}$. We infer

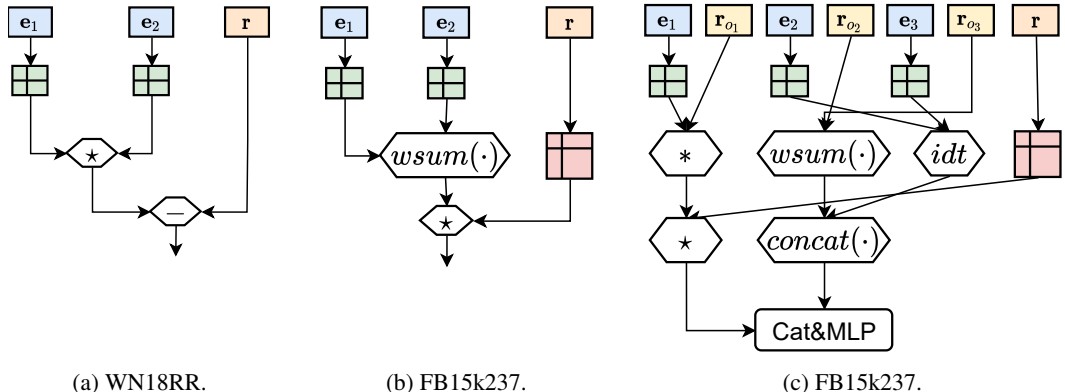

(a) WN18RR.          (b) FB15k237.          (c) FB15k237.

Figure 5: Several message functions searched by MSeaHKG on different data sets.

Table 8: The model comparison of the link prediction task on KGs. The results of R-GCN and CompGCN are copied from Vashishth et al. (2020), and others are copied from Rossi et al. (2021).

| type | model | FB15k237 | | | WN18RR | | |
|---|---|---|---|---|---|---|---|
| | | MRR | Hit@1 | Hit@10 | MRR | Hit@1 | Hit@10 |
| Geometric | TransE | 0.310 | 0.217 | 0.497 | 0.206 | 0.028 | 0.495 |
| | RotatE | 0.336 | 0.238 | 0.531 | 0.475 | 0.426 | 0.574 |
| Bilinear | DistMult | 0.313 | 0.224 | 0.490 | 0.433 | 0.397 | 0.502 |
| Tensor | TuckER | 0.352 | 0.259 | 0.536 | 0.459 | 0.430 | 0.514 |
| GNNs | R-GCN | 0.248 | 0.151 | 0.417 | - | - | - |
| | CompGCN | 0.355 | 0.264 | 0.535 | 0.479 | 0.443 | 0.546 |
| Search | MSeaHKG | **0.360** | **0.267** | **0.545** | **0.485** | **0.446** | **0.554** |

that is mainly because FB15k237 has 237 relations, which is more than 11 relations on WN18RR. For the message function searched on facts with high arity, we show an example on JF17K-3 in Fig. 5 (c). It demonstrates that the message function tends to be deeper and complex in the high arity case. Note that the full version of message functions searched on WikiPeople ($N = 9$) and JF17K ($N = 6$) is too large to put it on paper. Overall, we can observe that the message functions are data-dependent from Fig. 5.

### D.3 ABLATION STUDY

Except for main experimental results, here we report the performance of several variants of MSeaHKG (see Tab. 9) to investigate some key designs in this paper, including MSeaHKG$^{W_r}$, MSeaHKG$^{op}$, MSeaHKG$^{st}$ for the search space, MSeaHKG$^{darts}$ and MSeaHKG$^{rl}$ for the search algorithm.

#### D.3.1 SEARCH SPACE

We first present the configuration of variants:

- MSeaHKG$^{W_r}$ basically enables current GNN searching methods working on HKGs. Inspired by R-GCN (see Eq. 2), we first replace the transform matrix $W$ in $mg_c(\cdot)$ (see Eq. 1) to $W_r$. Then, we concatenate the entity embeddings as $h = concat(e_1, \ldots, e_n, \mathbf{0}, \ldots, \mathbf{0})$. Note that the number of zero embeddings $\mathbf{0}$ is equal to $n_{max} - n$. We utilize the message function $mg_c(r(e_1, \ldots, e_n)) = W_r h$ to replace Eq. 7. Other steps are same with original version.

- MSeaHKG$^{op}$ only searches operations of operators $\phi$ and $\gamma$ in $mg(\cdot; \boldsymbol{\theta})$ (i.e., $\boldsymbol{\theta} = \{\theta_i^{\phi_k}\} \cup \{\theta_i^{\gamma_k}\}$), while keeping the structure of StarE's message functions. Other steps are same with original version.

- MSeaHKG$^{st}$ only searches structures of the proposed message function $mg(\cdot; \boldsymbol{\theta})$, and sets $\phi, \gamma$ to $corr, wsum$ respectively (i.e., $\boldsymbol{\theta} = \{\theta_{ij}^k\}$). The fixed operations are selected based on better empirical performance. Other steps are same with original version.

Table 9: The comparison of variants of MSeaHKG in the link prediction task on HKGs.

| Type | Model | WikiPeople | | | | JF17K | | | |
|------|-------|------|-------|-------|--------|------|-------|-------|--------|
| | | MRR | Hit@1 | Hit@3 | Hit@10 | MRR | Hit@1 | Hit@3 | Hit@10 |
| GNNs | StarE | 0.378 | 0.265 | 0.452 | 0.542 | 0.542 | 0.454 | 0.580 | 0.685 |
| | G-MPNN | 0.367 | 0.258 | 0.439 | 0.526 | 0.530 | 0.459 | 0.572 | 0.688 |
| Search | S2S | 0.372 | 0.277 | 0.439 | 0.533 | 0.528 | 0.457 | 0.570 | 0.690 |
| | MSeaHKG | **0.395** | **0.291** | **0.470** | **0.554** | **0.577** | **0.481** | **0.599** | **0.711** |
| Variants of space | MSeaHKG$^{W_r}$ | 0.354 | 0.233 | 0.431 | 0.520 | 0.512 | 0.445 | 0.553 | 0.671 |
| | MSeaHKG$^{op}$ | 0.385 | 0.274 | 0.460 | 0.548 | 0.554 | 0.468 | 0.591 | 0.699 |
| | MSeaHKG$^{st}$ | 0.391 | 0.278 | 0.465 | 0.552 | 0.563 | 0.475 | 0.602 | 0.706 |
| Variants of algorithm | MSeaHKG$^{darts}$ | 0.373 | 0.275 | 0.445 | 0.535 | 0.554 | 0.460 | 0.588 | 0.697 |
| | MSeaHKG$^{rl}$ | 0.380 | 0.281 | 0.457 | 0.542 | 0.563 | 0.472 | 0.593 | 0.701 |

From Tab. 9, we observe that the simple extension version MSeaHKG$^{W_r}$ even cannot achieve as good performance as existing GNNs (e.g., StarE and G-MPNN). This verifies the claim that the simple message function in the existing GNN searching method (e.g., AutoGEL (Zhili et al., 2021) discussed in Sec. 2.2) may not be able to handle the complex correlations between relations and entities on HKGs. Moreover, MSeaHKG$^{op}$ keeps the same message function structure with StarE but searches suitable operations. Differ from manually tuning operations in StarE, the automatic way is more powerful so that MSeaHKG$^{op}$ achieves a minor improvement compared with StarE. As for MSeaHKG$^{st}$, it can search for more flexible structures of message functions for the given HKG and achieve the best performance among several variants. It can illustrate that the message function design is important to HKG embedding. However, MSeaHKG$^{st}$ is still slightly inferior compared with the original version of MSeaHKG. This shows that the best structure and operations are dependent. Simply fixing operations to search the structure may lead to the sub-optimum.

### D.3.2 SEARCH ALGORITHM

In this paper, we mainly focus on one-shot NAS search algorithms due to their searching efficiency (see more discussion in Sec. 2.1). Existing one-shot NAS algorithms can be roughly categorized into: stochastic differentiable method (e.g., SNAS (Xie et al., 2018)), deterministic differentiable method (e.g., DARTS (Liu et al., 2018)), and policy gradient-based method (e.g., ENAS (Pham et al., 2018), ASNG (Akimoto et al., 2019)). Here, we implement two more variants of MSeaHKG, MSeaHKG$^{darts}$ based on DARTS and MSeaHKG$^{rl}$ based on ASNG, to investigate the performance of other two kinds of NAS search algorithms in our application scenario.

- MSeaHKG$^{darts}$ follows DARTS to directly relax $\boldsymbol{\Theta}$ to learnable parameters. Then, the computation in DAG (see Eq. 6) will be reformed to:

$$\boldsymbol{h}_i^k = \sum_{j=1}^{|\mathcal{O}|} \alpha_{ij}^k \cdot o_i^k(\boldsymbol{h}_j^{k-1}), \ \alpha_{ij}^k = \frac{\theta_{ij}^k}{\sum_{j'=1}^{|\mathcal{O}|} \theta_{ij'}^k}. \tag{13}$$

Then, we are able to minimize the loss $\mathcal{L}(\boldsymbol{\Theta}, \boldsymbol{\omega}; \mathcal{G})$ through the gradient $\nabla_{\boldsymbol{\Theta}} \mathcal{L}(\cdot)$.

- MSeaHKG$^{rl}$ follows the policy gradient-based NAS search algorithm to derive:

$$\nabla_{\bar{\boldsymbol{\Theta}}} E_{X \sim p_{\bar{\boldsymbol{\Theta}}}(X)}[\mathcal{L}(X, \boldsymbol{\omega}; \mathcal{G})] = \nabla_{\bar{\boldsymbol{\Theta}}} \int p_{\bar{\boldsymbol{\Theta}}}(X)\mathcal{L}(X, \boldsymbol{\omega}; \mathcal{G})dX = \int \mathcal{L}(X, \boldsymbol{\omega}; \mathcal{G})\nabla_{\bar{\boldsymbol{\Theta}}} p_{\bar{\boldsymbol{\Theta}}}(X)dX$$

$$= \int \mathcal{L}(X, \boldsymbol{\omega}; \mathcal{G})\nabla_{\bar{\boldsymbol{\Theta}}} \log p_{\bar{\boldsymbol{\Theta}}}(X) \cdot p_{\bar{\boldsymbol{\Theta}}}(X)dX \tag{14}$$

$$= \nabla_{\bar{\boldsymbol{\Theta}}} E_{X \sim p_{\bar{\boldsymbol{\Theta}}}(X)}[\mathcal{L}(X, \boldsymbol{\omega}; \mathcal{G})\nabla_{\bar{\boldsymbol{\Theta}}} \log p_{\bar{\boldsymbol{\Theta}}}(X)],$$

where $\nabla_{\bar{\boldsymbol{\Theta}}} p_{\bar{\boldsymbol{\Theta}}}(X) = \log p_{\bar{\boldsymbol{\Theta}}}(X)p_{\bar{\boldsymbol{\Theta}}}(X)$ is the policy gradient trick (Williams, 1992). In practical, we utilize ASNG (Akimoto et al., 2019) to instantiate Eq. 14, which implements the fisher information matrix for fast convergence and adaptive learning rate for robustness.

We here analyze the model comparison between original MSeaHKG with its variants MSeaHKG$^{rl}$ and MSeaHKG$^{darts}$. From Tab. 9 and Tab. 10, we observe that MSeaHKG and MSeaHKG$^{rl}$ have

Table 10: The training time comparison (in hours) of GNN-based models in the link prediction task on HKGs.

| Type | Model | HKGs | | Arity=3 | | Arity=4 | |
|---|---|---|---|---|---|---|---|
| | | WikiPeople | JF17K | W-3 | J-3 | W-4 | J-4 |
| GNNs | G-MPNN | $15.4 \pm 2.6$ | $4.1 \pm 0.3$ | $1.2 \pm 0.1$ | $0.6 \pm 0.1$ | $0.5 \pm 0.1$ | $0.6 \pm 0.1$ |
| | StarE | $137.5 \pm 6.3$ | $16.7 \pm 2.2$ | $2.6 \pm 0.4$ | $4.7 \pm 0.8$ | $2.3 \pm 0.4$ | $1.9 \pm 0.2$ |
| Search | MSeaHKG | $30.9 \pm 4.7$ | $7.7 \pm 1.8$ | $1.8 \pm 0.4$ | $1.4 \pm 0.2$ | $0.9 \pm 0.1$ | $1.1 \pm 0.2$ |
| Variants of algorithm | MSeaHKG$^{darts}$ | $40.2 \pm 2.5$ | $11.5 \pm 2.2$ | $2.5 \pm 0.5$ | $2.1 \pm 0.3$ | $1.5 \pm 0.2$ | $1.3 \pm 0.1$ |
| | MSeaHKG$^{rl}$ | $35.1 \pm 3.0$ | $10.3 \pm 1.7$ | $2.3 \pm 0.3$ | $1.5 \pm 0.2$ | $1.4 \pm 0.2$ | $1.1 \pm 0.1$ |

better performance than MSeaHKG$^{darts}$ in both effectiveness and efficiency comparisons. First, DARTS aims to train a supernet by mixing all candidate operations during the searching phase, then it will derive a discrete architecture after finishing the search. But the weights $\boldsymbol{\alpha}_i^k$ in Eq. 13 cannot cannot converge to a one-hot vector, which lead to performance collapse after removing $|\mathcal{O} - 1|$ operations in $\mathcal{O}$ (Zela et al., 2019; Chu et al., 2020). Second, it will consume more computational resources when maintaining all operations during the search. Instead, MSeaHKG and MSeaHKG$^{rl}$ are to train discrete architectures in searching (the bounded discreteness of MSeaHKG is discussed in Appx. C.4), which avoids performance collapse and large computational overhead. As for the comparison between MSeaHKG with MSeaHKG$^{rl}$, MSeaHKG achieves slight improvements. That is mainly because MSeaHKG directly calculates the gradient w.r.t. $\bar{\Theta}$ from the loss $\mathcal{L}(\cdot)$, while MSeaHKG$^{rl}$ takes the loss $\mathcal{L}(\cdot)$ as a reward to feed it to a RL controller.

From Tab. 10, we also can observe the efficiency comparison between GNN-based models (G-MPNN, StarE, and MSeaHKG). Like other NAS methods, MSeaHKG requires two training phases, searching architecture and training the searched architecture from the scratch. Thus, it needs more running time compared with G-MPNN. Moreover, both G-MPNN and MSeaHKG utilizes the simple decoders (i.e., scoring function discussed in Appx. C.2), while StarE adopts the complex transformer as its decoder. Thus, StarE is less efficient.

### D.4 MSEAHKG CAN BE TRANSFERRED TO OTHER GRAPH-BASED TASKS.

Since many configurations of MSeaHKG are inspired by GNNs and GNN searching methods, we transfer MSeaHKG to other GNN-related tasks to further investigate its capability.

First, we conduct an extensive experiment on social recommendation (Fan et al., 2019), where the recommendation data sets are formed as multi-relational graphs. In the data set Ciao [2], nodes represent the users and items, edges have two main types: 1) 5 level of ratings $\{1, 2, 3, 4, 5\}$ between users and items, 2) connections among users. The goal of the task is to predict the unknown ratings of items given by users. After treating the 5 ratings as 5 edge types, the recommendation task is converted to the relation prediction task, i.e., predict the rating $r$ given $?(user, item)$. To compare MSeaHKG with literature more conveniently, we follow Fan et al. (2019) to utilize the Mean Absolute Error (MAE) and Root Mean Square Error (RMSE) as the evaluation metrics, and 60% as training data. We let MSeaHKG compute top-3 scores like $a = score(2(user, item)), b = score(4(user, item)), c = score(5(user, item))$, then adopt weighted sum to output the final rating like $(a \cdot 2 + b \cdot 4 + c \cdot 5)/(a + b + c)$. The experimental report in Tab. 11a shows MSeaHKG has a good generalized ability to the social recommendation task, which is consistent with the performance on KGs (i.e., multi-relational graphs).

Second, we extend MSeaHKG to graph-level tasks. We incorporate one more essential component at the final layer of MPNNs, i.e., readout funcion $rd(\cdot)$. General MPNNs employ $rd(\cdot)$ to output the representation of a whole graph $\mathcal{G}(E, R, S)$ by aggregating the node embeddings as:

$$h_\mathcal{G} = rd(\{e_i\}_{e_i \in E}).$$

Due to the modeling of edge representations, MSeaHKG slightly adjusts the readout function as:

$$h_G = rd(\{[e_1, r, e_2]\}_{r(e_1, e_2) \in S}).$$

---

[2]https://www.cse.msu.edu/~tangjili/datasetcode/truststudy.htm

Table 11: The model performance of extending MSeaHKG to other tasks.

(a) The comparison on the rating prediction task of social recommendation. The results of baselines are copied from Fan et al. (2019).

| model | Ciao | |
|---|---|---|
| | MAE | RMSE |
| PMF | 0.9520 | 1.1967 |
| TrustMF | 0.7681 | 1.0543 |
| NeuMF | 0.8251 | 1.0824 |
| GraphRec | 0.7540 | 1.0093 |
| MSeaHKG | 0.7511 | 1.0021 |

(b) The comparison on the graph classification task. The results of baselines on PROTEINS and IMDB-M are copied from PAS (Wei et al., 2021), and those on MUTAG are copied from GIN (Xu et al., 2018).

| type | model | PROTEINS | IMDB-M | MUTAG |
|---|---|---|---|---|
| **GNNs** | GCN | 0.7484 | 0.5040 | 0.8560 |
| | GraphSAGE | 0.7375 | 0.4853 | 0.8510 |
| | GIN | 0.7620 | 0.5230 | 0.8940 |
| **NAS for GNNs** | GraphNAS | 0.7520 | 0.4827 | - |
| | SNAG | 0.7233 | 0.5000 | - |
| | You et al. (2020) | 0.7390 | 0.4780 | - |
| | PAS | 0.7664 | 0.5220 | - |
| | MSeaHKG | 0.7724 | 0.5317 | 0.8922 |

We implement the choices of readout function as $\{global\_mean, global\_max, global\_sum\}$. We report the accuracy in Tab. 11b. Among 3 data sets, the graphs in MUTAG are multi-relational, thus most of GNN searching methods do not include it in empirical study. We can observe that MSeaHKG achieves not bad performance on these data sets.

