# OpenReview forum: "Message Function Search for Hyper-relational Knowledge Graph"
_ICLR.cc/2022/Conference — ICLR 2022 Submitted_

### Official Review · Reviewer_y4dP · 2021-10-25

**Correctness:** 3
**Technical Novelty And Significance:** 2
**Empirical Novelty And Significance:** 3
**Recommendation:** 5
**Confidence:** 4

**Main Review:**

Overall I feel the paper is nicely written and easy to follow. I am slightly confused by the main point of the paper. Is the main contribution of the paper proposing a new HKG method that achieves state-of-the-art in link prediction and relation prediction over HKGs, or the main contribution is a NAS platform with different configurations of GNNs in the search space? It seems that the authors claim the contribution to be the second but in the experiment, they just show that here’s a new method that achieves SOTA compared with prior methods. I am mostly curious about the performance of other prior baselines after doing the same hyperparameter search using the proposed framework. Besides, aren’t all the methods and details in Sec 3.2 already well established in the literature, including but not limited to how to frame the optimization as a search problem, gumbel softmax, etc.

- “But in scenarios of HKGs, it is important to know which relation (edge type) makes entities (nodes) adjacent.” What do you mean by this sentence?
- It should be “LP/RP” instead of “LR/RP” in the last row of Table 1.
- “firs” -> “first”, ‘cases studies” -> “case studies”
- The paper mentioned that they relax the discrete search space into a continuous and probabilistic space. How did you relax discrete parameters like different aggregation and activation functions?
- What’s the difference between the position matrix and role embedding? I assume both are used to distinguish entities at different positions in the hyper edge.
- Many of the configurations in Sec. 3 seem to be also applicable in general graph neural networks, e.g, concatenation, composition, residual connection and so on. It would be helpful to clarify which ones are specific to HKG, which ones are not. Otherwise it’s confusing and it’s unclear how novel the proposed design choices are.


**Summary Of The Paper:**

The paper proposes a new neural architecture search framework over graph neural networks in hyper relational knowledge graphs (HKG). The main motivation is that the current GraphNAS system does not have an emphasis on edge representations and how message passing functions handle position-sensitive information on HKG. The paper proposes a new search space of these configurations and proposes a new HKG method that achieves state-of-the-art on several benchmarks.

**Summary Of The Review:**

I feel the paper is not currently clear in the contribution and there are some disjoint between the method section and the experiment section. Also I am not entirely convinced of the necessity of a new NAS framework specifically designed for GNNs on HKGs. It seems like a problem of limited scope, and many new configurations also apply to GNNs in other tasks.

---

> ### Author Response · Authors · 2021-11-21
> **We replied to the concerns raised by the reviewer y4dP.**
>
> We sincerely appreciate the valuable and constructive comments from the reviewers, and our detailed replies are as follows.
>
> ## Replies to Some Concerns
> Here we summarize several concerns from the main review and summary of the review as below. Please correct us if we misunderstand the meaning of any reviews.
>
> **Ry4dP-C1:** I feel the paper is not currently clear in the contribution and there are some disjoint between the method section and the experiment section.
> * I am slightly confused by the main point of the paper. Is the main contribution of the paper proposing a new HKG method that achieves state-of-the-art in link prediction and relation prediction over HKGs, or the main contribution is a NAS platform with different configurations of GNNs in the search space? It seems that the authors claim the contribution to be the second but in the experiment, they just show that here’s a new method that achieves SOTA compared with prior methods.
> * Besides, aren’t all the methods and details in Sec 3.2 already well established in the literature, including but not limited to how to frame the optimization as a search problem, gumbel softmax, etc.
>
> **Reply to Ry4dP-C1:** We first further clarify the claims and technical contributions presented in the previous submission.
>
> * For contributions in the previous submission (see the end of Sec. 1), we claimed that "In this paper, we propose a searching method to dynamically design a suitable GNN that can achieve high performance on the given HKG." We summarized the main contribution is to propose a GNN searching method MSeaHKG specifically designed for HKGs. Thus, the experiments mainly demonstrate the effectiveness of MSeaHKG on HKGs tasks. Besides, most of the existing NAS for GNNs methods are not designed for multi-relational graphs/hypergraphs. They cannot learn the relational representations, thus fail to handle tasks on KGs/HKGs. To compare with NAS for GNNs methods, we extend another GNN searching framework AutoGEL from KGs to HKGs (named $\text{MSeaHKG}^{W_r}$) as shown in Tab. 9 of the submission.
>
> * Yes, the literature has established the optimization adopted in this paper. Thus, we did not claim it as our contribution (see end of Sec. 1). We just presented the overall optimization procedure in a stochastic differentiable way.
>
> We have realized that several sentences and statements may lead to misunderstanding. Thus, we have revised them in Sec.1. Besides, as suggested by reviewer WuXj and **Ry4dP-C3**, we enlarge the scope of this paper and extend MSeaHKG to other GNN-related tasks. Please mainly refer to **Ry4dP-C3**.
>
> ========
>
> **Ry4dP-C2:** I am mostly curious about the performance of other prior baselines after doing the same hyperparameter search using the proposed framework.
>
> **Reply to Ry4dP-C2:** Most of the baselines reported in the previous submission have already been tuned with some hyperparameter optimization packages, especially those works with competitive results.
> * For GNNs on HKGs (StarE and G-MPNN in Tab. 2,3,4), we re-implement them due to the different experimental settings (check **RgMZD-D1** for more details) and tune them with the help of package optuna.
> * The searching method S2S (in Tab. 2,4) has been tuned with the TPE sampler based on the implementation HyperOpt https://github.com/hyperopt/hyperopt (see S2S paper).
> * For tensor models (n-CP, n-TuckER, GETD in Tab. 4), they are tuned with the help of optuna (see GETD paper).
>
> ========
>
> **Ry4dP-C3:** Also I am not entirely convinced of the necessity of a new NAS framework specifically designed for GNNs on HKGs. It seems like a problem of limited scope, and many new configurations also apply to GNNs in other tasks.
>
> **Reply to Ry4dP-C3:** Thanks for the suggestions. In the previous submission, we mainly conduct experimental results on edge-level tasks (link/relation prediction) on multi-relational hypergraphs (HKGs). As suggested by Reviewers WuXj and y4dP, we extend MSeaHKG to more variety of tasks and graphs since many configurations of MSeaHKG can be applied to GNNs in other tasks. Please mainly refer to **Reply to RWuXj** due to the space limit.

---

> > ### Author Response · Authors · 2021-11-21
> > **We replied to detailed comments raised by the reviewer y4dP.**
> >
> > ## Replies to Detailed Comments
> >
> > **Ry4dP-D1:** The meaning of the statement "But in scenarios of HKGs, it is important to know which relation (edge type) makes entities (nodes) adjacent" is unclear.
> >
> > **Reply to Ry4dP-D1:** The classic MPNNs focus on the neighborhood information to propogate the node embeddings, such as $e_i = act(\sum_{e_j\in N(e_i)}We_j)$ (Eq. 1). In the scenarios of either KGs or HKGs, the edges between different entities (i.e., nodes) are labeled by various relations. In other words, such scenario requires the MPNN models to capture one more information resouce, i.e., what is type of edge that connects several entities? For example, one prior work CompGCN uses $e_i = act(\sum_{r(e_i,e_j)\in I(e_i)}\phi(e_j))$ (Eq. 3) to learn relational representations.
> >
> > We have realized that the previous statement may casue some misunderstanding and corrected it in Sec. 1.
> >
> > ========
> >
> > **Ry4dP-D2-D3:** Several typos exist in the submission, such as "LR/RP", "firs", and "cases studies".
> >
> > **Reply to Ry4dP-D2-D3:** Thank you for pointing out the typos. We have corrected them and polished the paper.
> >
> > ========
> >
> > **Ry4dP-D4:** The paper mentioned that they relax the discrete search space into a continuous and probabilistic space. How did you relax discrete parameters like different aggregation and activation functions?
> >
> > **Reply to Ry4dP-D4:** Generally, the relaxation of other functions (e.g., aggregation and activation) is consistent with $\phi^k(h) = \sum \theta^{\phi_k}_i o_i(h)$ and $\gamma^k(h) = \sum \theta^{\gamma_k}_i o_i(h)$ as presented in Sec. 3.1. Here we illustrate more details.
> >
> > * Aggregation function $agg(\cdot)$: $m_i = \sum_{o_j\in \mathcal{O}^{ag}} \theta_j^{ag} \cdot o_j (\\{mg(r(e_1,\dots,e_n);\theta^{mg})\\}_{r(e_1,\dots,e_n)\in I(e_i)})$, where $mg(r(e_1,\dots,e_n);\theta^{mg})$ is outputted from Eq. 7 and $\mathcal{O}^{ag}=\\{sum,mean,max\\}$ (See Sec. 2.1).
> >
> > * Activation function $act(\cdot)$: $e_i = \sum_{o_j\in \mathcal{O}^{at}} \theta_j^{at} \cdot o_j(comb(e_i, m_i))$, where $\mathcal{O}^{at}=\\{identity,sigmoid,tanh\\}$.
> >
> > Then, as presented in Sec. 3.2, we can relax $\theta^{ag}$ and $\theta^{at}$ into continuous space $\bar{\theta}^{ag}$ and $\bar{\theta}^{at}$ by leveraging the Gumbel-Softmax technique.
> >
> > We have realized that the previous explanation is not sufficient. Thus, we present more details in Appx. C.1.
> >
> > ========
> >
> > **Ry4dP-D5:** What is the difference between the position matrix and role embedding? I assume both are used to distinguish entities at different positions in the hyper edge.
> >
> > **Reply to Ry4dP-D5:** Overall, positional matrices encode the positional information of entities in hyper-relational facts, while role embeddings are utilized to model the semantic information of entities. Usually, the role represents the hypernym of entities.
> >
> > Here we raise two examples to further clarify their difference. For example, a 3-ary fact playCharacerIn(actor:ZacharyQuinto,character:Spock,movie:StarTrek) is composed of the relation playCharacerIn, roles {actor,character,movie} and entities {ZacharyQuinto,Spock,StarTrek}.
> > For another fact, actorAwardIn(actor:ZacharyQuinto,award:bestCastBSFC,movie:StarTrek). We may easily observe that the entities Spock and bestCastBSFC are located in the 2nd position of the 3-ary facts. But their semantical meanings are quite different, where one is character and another is the award. Besides, as presented in Sec. 2.1 StarE explicitly requires more inputs to model the role $r_{o_j}$ compared with G-MPNN, which also indicate that the roles and positions can be distinguished. Thus, we propose the positional matrix $W_{P(e)}$ and role embedding $r_{o}$ to encode these information sources, respectively.
> >
> > Thanks for reminding us. We have clarified it in Sec. 3.1.

---

> > > ### Author Response · Authors · 2021-11-21
> > > **We replied to the rest comment raised by the reviewer y4dP**
> > >
> > > **Ry4dP-D6:** Many of the configurations in Sec. 3 seem to be also applicable in general graph neural. It would be helpful to clarify which ones are specific to HKG, which ones are not. Otherwise, it’s confusing and it’s unclear how novel the proposed design choices are.
> > >
> > > **Reply to Ry4dP-D6:** Here we clarify the operation choices in the message function (i.e., $\mathcal{O}$ in Sec. 3.1) that is specifically designed for HKGs. First, we utilize the $W_{P(e)}$ and $P(e): e\rightarrow \\{1,\dots,N\\}$ to capture the more complex and high-order positions in hype-relational facts, while at most 2 positions in classic graphs. Second, except for node and relation embeddings, MSeaHKG proposes more representations $r_o$ to encode the role information.
> > >
> > > Please note that the designs of $\mathcal{O}$ are mainly inspired by existing designs of classic GNNs or GNN applications. Thus, these configurations allow us to extend the proposed MSeaHKG to other tasks and kinds of graphs. We have discussed more extensive experiments in **Reply to RWuXj**. Besides, we would like to further clarify the major novelty of the proposed search space is the message function space design instead of the chosen design. In this submission, we investigate the functionality of GNNs' message functions in the HKG area and propose a novel space for message functions. Our space follows the principle of NAS space designs. It can represent some representative works as special cases, including GNNs on HKGs (CompGCN, StarE, G-MPNN in Fig. 1) and classic KGE models (e.g., HolE, RotatE, n-CP, and n-DistMult in Fig. 4).

---

### Official Review · Reviewer_gMZD · 2021-11-01

**Correctness:** 4
**Technical Novelty And Significance:** 3
**Empirical Novelty And Significance:** 3
**Recommendation:** 6
**Confidence:** 3

**Main Review:**

Pros:
* Automatically searching message-passing functions for HKGs is an important yet underexplored problem.
* The proposed search space seems to incorporate most state-of-the-art MPNNs for HKGs.
* The experimental results are comprehensive, with ablation studies, examples, and various analyses provided in the appendix.

Concerns and questions:
* The writing of the paper could be improved, especially the methodology part where there are many symbols involved, which is hard to follow.
* Some experimental results are not consistent with results presented in previous papers (e.g., StarE and G-MPNN). The authors need to explain where the difference comes from.
* The code is not available yet, so there is currently no way to assess the proposed method experimentally. I believe the authors would make the code publicly available at publication time?


**Summary Of The Paper:**

In this paper, the authors propose to conduct neural architecture search (NAS) for hyper-relational knowledge graphs (HKGs). Compared with normal graphs, HKGs can better model the complex relationships between different entities. Specifically, a novel search space is proposed inspired by recent message-passing GNNs on HKGs such as G-MPNN, CompGCN, StarE, etc., and a differentiable search algorithm is utilized. Experiments demonstrate the effectiveness of the proposed method.

**Summary Of The Review:**

Overall, I think this paper makes non-trivial contributions in the search space design for HKGs with extensive experiments, and thus may be interesting to both KGs and NAS community.

===after rebuttal===
I have read the rebuttal and my score remains positive.

---

> ### Author Response · Authors · 2021-11-21
> **We replied to comments raised by the reviewer gMZD, including more discussions and more experimental reports.**
>
> We sincerely appreciate the valuable and constructive comments from the reviewers, and our detailed replies are as follows.
>
> ## Replies to the Concern
> **RgMZD-C1:** The writing of the paper could be improved, especially the methodology part where there are many symbols involved, which is hard to follow.
>
> **Reply to RgMZD-C1:** Thanks for the suggestions. Because of complex symbols, we assume that Eq. (5) and (6) are the most difficult parts to follow. Unfortunately, we cannot simply them anymore. For example, we have to keep superscripts and subscripts $\\{i,j,k\\}$ to denote the current operation, the previous operation and current layer of message functions.
>
> Instead, we have revised the Fig. (c) to illustrate the Eq. (5) and (6) more clearly.
>
>
> ## Replies to Detailed Comments
>
> **RgMZD-D1:** Some experimental results are not consistent with results presented in previous papers (e.g., StarE and G-MPNN). The authors need to explain where the difference comes from.
>
> **Reply to RgMZD-D1:**
> In the previous submission (Sec. 4.1), we simply mention that "we compare models in the transductive settings" at task part and "G-MPNN and StarE (we re-implement G-MPNN and StarE because their initial task scenarios are not exactly match tasks)" at the baseline part of Sec. 4.1. Thanks for reminding us. Now we realize that such explanation is not enough. We have revised the statement in Sec. 4.1 and added the sources of results in the captions of tables.
>
> G-MPNN mainly conducts experiments on the inductive setting, i.e., test entities are unseen in the training phase. In this paper, we follow the default transductive setting in KGs or HKGs [1][2]. Thus, we reimplement G-MPNN and present a higher performance than the original reports. As for StarE, it only shows the results of link prediction on the sro triplets of hyper-relational facts. Intuitively, given a hyper-relational fact $r(e_1,e_2,...,e_n)$, StarE only test the performance on two positions (i.e., $r(?,e_2,...,e_n)$ and $r(e_1,?,...,e_n)$), while the proposed MSeaHKG tests the performance on $n$ positions as existing works [3][4].
>
> Besides, here we report MSeaHKG on their original settings for a more fair comparison. As shown in below tables, MSeaHKG still achieves better performance in most evaluation measurements.
>
> |model|WIKI-People(MRR/H@1/H@3)|JF17K(MRR/H@1/H@3)|
> |:----:|:----:|:----:|
> |G-MPNN-sum|0.287/0.216/0.308|0.439/0.395/0.461|
> |G-MPNN-mean|0.281/0.220/0.315|0.442/0.391/0.460|
> |G-MPNN-max|0.289/0.218/0.310|0.446/0.393/0.464|
> |MSeaHKG|0.305/0.225/0.331|0.458/0.402/0.481|
>
> |model|WIKI-People(MRR/H@1/H@5/H@10)|JF17K(MRR/H@1/H@5/H@10)|
> |:----:|:----:|:----:|
> |StarE (H)|0.491/0.398/0.592/0.648|0.574/0.496/0.658/0.725|
> |StarE (T)|0.493/0.400/0.592/0.648|0.562/0.493/0.637/0.702|
> |MSeaHKG|0.496/0.411/0.603/0.634|0.593/0.515/0.688/0.742|
>
> ========
>
> **RgMZD-D2:** The code is not available yet, so there is currently no way to assess the proposed method experimentally. I believe the authors would make the code publicly available at publication time?
>
> **Reply to RgMZD-D2:** We will make the code available when the paper is public.
>
>
> ## References
> [1] Knowledge Graph Embedding: A Survey of Approaches and Applications, TKDE 2017.
>
> [2] Knowledge Graph Embedding for Link Prediction: A Comparative Analysis, TKDD 2021.
>
> [3] Generalizing Tensor Decomposition for N-ary Relational Knowledge Bases, WWW 2020.
>
> [4] Role-Aware Modeling for N-ary Relational Knowledge Bases, WWW 2021.

---

### Official Review · Reviewer_WuXj · 2021-11-02

**Correctness:** 4
**Technical Novelty And Significance:** 3
**Empirical Novelty And Significance:** 3
**Recommendation:** 6
**Confidence:** 3

**Main Review:**

I think the work is novel in including the message function in the search space for NAS. This introduced a novel way to learn the representation using flexibility of the knowledge from the message functions. The search algorithm is a variation of the NAS tailored for this framework.

The experiments shows better results for the proposed method in comparison with other baselines. The other methods shows variable effectiveness over different datasets, but the proposed model beats all of them in all the presented results.

The ablation study also looks good and shows effect of different message passing functions and algorithms for the framework.

We have result on different arity (3 and 4) for the same dataset. It would be interesting to see the relation between arity and effectiveness of the method tabulated.

Authors have shown result in link prediction task and relation prediction tasks. It would be interesting to see the performance in more variety of tasks.

It may be outside of the scope of this paper, but one question can be how should the framework perform in non HKG problems? May be we can get good result for other one shot problems in graph.







**Summary Of The Paper:**


In this paper, authors propose a new message function searching method for Hyper Relational Knowledge Graphs. They also proposed a search space of message functions. Finally the authors proposed a one shot NAS algorithm for searching in the message function space and also over other GNN components for a given Hyper Relational Knowledge Graph.




**Summary Of The Review:**

Overall I think the paper is technically strong and authors have shown effectiveness of their method. Although the scope may be somehow limited.

---

> ### Author Response · Authors · 2021-11-21
> **We have added more experimental reports to enhance the scope of this paper as suggested by the reviewer WuXj.**
>
> We sincerely appreciate the valuable and constructive comments from the reviewers, and our detailed replies are as follows.
>
> ## Replies to Summary of the Review
> **RWuXj:** The scope may be somehow limited.
> * Authors have shown result in link prediction task and relation prediction tasks. It would be interesting to see the performance in more variety of tasks.
> * It may be outside of the scope of this paper, but one question can be how should the framework perform in non HKG problems? May be we can get good result for other one shot problems in graph.
>
> **Reply to RWuXj:** Thanks for the suggestions. In the previous submission, we mainly conduct experimental results on edge-level tasks (link/relation prediction) on multi-relational hypergraphs (HKGs). As suggested by Reviewers WuXj and y4dP, we extend MSeaHKG to more variety of tasks and graphs since many configurations of MSeaHKG can be applied to GNNs in other tasks.
>
> **more edge-level tasks on multi-relational graphs:** We first compare the link prediction task on WN18RR and FB15k237 to check the capability of MSeaHKG on classic KGs. As shown in the below table, MSeaHKG achieves good performance on KGs, which is consistent with HKGs.
>
> |type|model|FB15k237(MRR/Hit@10)|WN18RR(MRR/Hit@10)|
> |:----:|:----:|:----:|:----:|
> |Translating|TransE|0.310/0.497|0.206/0.495|
> ||RotatE|0.336/0.531|0.475/0.574|
> |Tensor|TuckER|0.352/0.536|0.459/0.514|
> |GNNs|R-GCN|0.248/0.417|-|
> ||CompGCN|0.355/0.535|0.479/0.546|
> |Search|MSeaHKG|0.360/0.545|0.485/0.554|
>
> Then, we conduct another extensive experiment on social recommendation, where the recommendation data sets are formed as multi-relational graphs. In the data set Ciao (https://www.cse.msu.edu/~tangjili/datasetcode/truststudy.htm), nodes represent the users and items, edges have two main types: 1) 5 level of ratings $\\{1,2,3,4,5\\}$ between users and items, 2) connections among users. The goal of the task is to predict the unknown ratings of items given by users. After treating the 5 ratings as 5 edge types, the recommendation task is converted to the relation prediction task, i.e., predict the rating $r$ given $?(user,item)$. To compare MSeaHKG with literature more conveniently, we follow [1] to utilize the Mean Absolute Error (MAE) and Root Mean Square Error (RMSE) as the evaluation metrics, and 60\% as training data. Thus, we let MSeaHKG compute top-3 scores like $a=score(2(user,item)), b=score(4(user,item)), c=score(5(user,item))$, then adopt weighted sum to output the final rating like $(a\cdot 2+b\cdot 4+c\cdot 5)/(a+b+c)$. The experimental report shows that MSeaHKG can also be transferred to other graph-based tasks.
>
> |model|MAE|RMSE|
> |:----:|:----:|:----:|
> |Probabilistic MF|0.952|1.1967|
> |TrustMF|0.7681|1.0543|
> |NeuMF|0.8251|1.0824|
> |GraphRec (GNN) [1]|0.7540|1.0093|
> |MSeaHKG|0.7511|1.0021|
>
> Generally, MSeaHKG focuses on learning better edge representations, thus it achieves outstanding performance on edge-level tasks. Besides, we adapt it to the graph classification task to see more performance comparison.
>
> **graph classificaion task:** To adapt MSeaHKG to graph-level tasks, we incorporate one more essential component at the final layer of MPNNs, i.e., readout funcion $rd(\cdot)$. General MPNNs employ $rd(\cdot)$ to output the representation of a whole graph $\mathcal{G}(E,R,S)$ by aggregating the node embeddings as: $h_{G} = rd(\\{e_i\\}_{e_i \in E})$.
>
> Due to the modeling of edge representations, MSeaHKG slightly adjusts the readout function as: $h_{G} = rd(\\{[e_1,r,e_2]\\}_{r(e_1,e_2)\in S})$. We implement the choices of readout function as $\\{global\\_mean,global\\_max,global\\_sum\\}$. We report the accuracy as follows. Among 3 data sets, the graphs in MUTAG are multi-relational, thus most of GNN searching methods do not include it in empirical study. We can observe that MSeaHKG achieves not bad performance on these data sets.
>
> |type|model|PROTEINS|IMDB-M|MUTAG|
> |:----:|:----:|:----:|:----:|:----:|
> |GNNs|GCN|0.7484|0.5040|0.8560|
> ||GraphSAGE|0.7375|0.4853|0.8510|
> ||GIN|0.7620|0.5230|0.8940|
> |NAS for GNNs|GraphNAS|0.7520|0.4827|-|
> ||SNAG|0.7233|0.5000|-|
> ||You 2020|0.7390|0.4780|-|
> ||PAS|0.7664|0.5220|-|
> ||MSeaHKG|0.7724|0.5317|0.8922|
>
> Note that extensive experiments are conducted on parts of benchmark data sets due to limited time in rebuttal. We have included above experimental reports in Tab. 8 and 11, and added Appendix D.4.
>
>
> ## References
> [1] Graph Neural Networks for Social Recommendation, WWW 2019.

---

> > ### Author Response · Authors · 2021-11-21
> > **More Replies**
> >
> > Due to the space limit of 5000 characters, we reply another suggestion here.
> >
> > ## Replies to Detailed Comments
> >
> > **RWuXj-D1:** We have result on different arity (3 and 4) for the same dataset. It would be interesting to see the relation between arity and effectiveness of the method tabulated.
> >
> > **Reply to RWuXj-D1:** We simply mentioned the correlation in the last paragraph of Sec. 4.2. To investigate more about the correlation, we add more experiments of GNNs on the data sets with a fixed arity. Compared with GNNs, the tensor-based models (e.g., GETD and S2S) tend to achieve higher performance in the fixed case. And S2S is slightly inferior to GNNs in the mixed case. It corresponds to our claim about the weakness of S2S. Besides, MSeaHKG works well in both cases.
> >
> > |type|model|WP|WP-3|WP-4|JF|JF-3|JF-4|
> > |:----:|:----:|:----:|:----:|:----:|:----:|:----:|:----:|
> > |Tensor|n-TuckER|-|0.365|0.362|-|0.727|0.804|
> > ||GETD|-|0.373|0.386|-|0.732|0.810|
> > |GNNs|StarE|0.378|0.369|0.382|0.542|0.728|0.811|
> > ||G-MPNN|0.367|0.366|0.375|0.530|0.730|0.802|
> > |Search|S2S|0.372|0.386|0.391|0.528|0.740|0.822|
> > ||MSeaHKG|0.395|0.405|0.412|0.577|0.757|0.834|

---

### Decision · Program_Chairs · 2022-01-20

**Decision:**

Reject

**Comment:**

The paper studies  neural architecture search for hyper-relational knowledge graphs (HKGs).  A  search space is put-forth, and  it is  searched with a differentiable search algorithm. The paper is technically strong. However,  there are some concerns about the narrow scope of the problem/solution, given that other more general formulations have also been studied.